# A PRIMAL-DUAL ALGORITHM FOR VARIATIONAL IMAGE RECONSTRUCTION WITH LEARNED CONVEX REGULARIZERS

## ABSTRACT

We address the optimization problem in a data-driven variational reconstruction framework, where the regularizer is parameterized by an input-convex neural network (ICNN). While gradient-based methods are commonly used to solve such problems, they struggle to effectively handle non-smoothness which often leads to slow convergence. Moreover, the nested structure of the neural network complicates the application of standard non-smooth optimization techniques, such as proximal algorithms. To overcome these challenges, we reformulate the problem and eliminate the network's nested structure. By relating this reformulation to epigraphical projections of the activation functions, we transform the problem into a convex optimization problem that can be efficiently solved using a primal-dual algorithm. We also prove that this reformulation is equivalent to the original variational problem. Through experiments on several imaging tasks, we demonstrate that the proposed approach outperforms subgradient methods in terms of both speed and stability.

## 1 INTRODUCTION

Image restoration focuses on reconstructing high-quality images from degraded, low-quality versions that often result from issues during image acquisition and transmission. This includes tasks such as image denoising, image deblurring, image inpainting and computer tomography (CT) reconstruction. The measurement process is typically modeled as $\mathbf{y} = \mathbf{A}\mathbf{x} + \boldsymbol{\epsilon}$, where $A$ simulates the physics in the measurement process and $\boldsymbol{\epsilon}$ denotes the measurement noise. One then seeks to recover the unknown image $\mathbf{x}$ from the noisy measurement $\mathbf{y}$. To mitigate the ill-possedness of the inverse problem, the classical variational reconstruction framework incorporates prior information about plausible reconstructions through a regularizer:

$$\min_{\mathbf{x}} D(\mathbf{A}\mathbf{x}, \mathbf{y}) + \gamma R_{\boldsymbol{\theta}}(\mathbf{x}), \tag{P}$$

where $D$ is the data fidelity. The regularizer $R_{\boldsymbol{\theta}}$ can be parametrized, with $\theta$ denoting its parameters. The trade-off between data fidelity and regularizer is controlled by the positive regularization parameter $\gamma$. The reconstruction is obtained by solving the minimization problem (P).

Traditional methods often utilize hand-crafted regularizers, such as total variation (TV) (Rudin et al., 1992), total generalized variation (TGV) (Bredies et al., 2010) and sparsity promoting regularizer (Daubechies et al., 2004). In recent years, data-driven approaches for inverse problems have gained increasing interest. For instance, (Chen et al., 2017; Jin et al., 2017; Kang et al., 2017) propose learning end-to-end neural networks to post-process analytical reconstructions. Another prominent strategy involves unrolling methods (Adler & Öktem, 2018; Kobler et al., 2017; Meinhardt et al., 2017; Yang et al., 2016), which integrate neural network modules into iterative optimization algorithms based on the variational framework. Alternatively, several works (Aharon et al., 2006; Chen et al., 2014; Kunisch & Pock, 2013; Xu et al., 2012) attempted to learn regularizers. This is also extended to parameterizing them with neural networks (Goujon et al., 2023; Kobler et al., 2020; Li et al., 2020; Lunz et al., 2018; Mukherjee et al., 2020), and embedding them within variational reconstruction frameworks.

## 1.1 LEARNED CONVEX REGULARIZER WITH ICNNS

In Amos et al. (2017), a $L$-layered ICNN is defined by the following architecture:

$$
\begin{aligned}
\mathbf{z}_1 &= h_1(\mathbf{V}_0\mathbf{x} + \mathbf{b}_0), \\
\mathbf{z}_{i+1} &= h_{i+1}(\mathbf{V}_i\mathbf{x} + \mathbf{W}_i\mathbf{z}_i + \mathbf{b}_i),\ i = 1, \ldots, L-2, \\
R_{\boldsymbol{\theta}}(\mathbf{x}) &:= h_L(\mathbf{V}_{L-1}\mathbf{x} + \mathbf{W}_{L-1}\mathbf{z}_{L-1} + \mathbf{b}_{L-1}),
\end{aligned}
\tag{EQ}
$$

where $\mathbf{V}_i, \mathbf{W}_i$ are linear operators, which could represent various neural network components, such as fully connected layers, convolution layers and average pooling layers. Here $\boldsymbol{\theta} = \{\mathbf{V}_i, \mathbf{W}_i, \mathbf{b}_i\}$ represents the collection of all trainable parameters of the ICNN. The functions $h_i$ are non-linear activations. For $\mathbf{x}, \mathbf{y} \in \mathbb{R}^n$, we denote $\mathbf{x} \leq \mathbf{y}$ if $\mathbf{x}_i \leq \mathbf{y}_i$ for $i = 1, \ldots, n$. To handle general activations, we call a function $f : \mathbb{R}^n \to \mathbb{R}^m$ convex if $f(\alpha\mathbf{x} + (1-\alpha)\mathbf{y}) \leq \alpha f(\mathbf{x}) + (1-\alpha)f(\mathbf{y})$ for every $\mathbf{x}, \mathbf{y} \in \mathbb{R}^n$ and $\alpha \in [0, 1]$. $f$ is called non-decreasing if $f(\mathbf{x}) \leq f(\mathbf{y})$ for $\mathbf{x} \leq \mathbf{y}$.

The convexity of $R_{\boldsymbol{\theta}}$ with respect to the input $\mathbf{x}$ can be guaranteed by imposing that the weights $\mathbf{W}_i$ are non-negative and $h_i$ are convex, non-decreasing.

A major advantage of a convex setting over a non-convex one is the ability to compute a global optimum independent of initialization, allowing one to leverage the well-established theory of convex optimization with guaranteed convergence to efficiently solve (P). Therefore, we focus on the case where the regularizer $R_{\boldsymbol{\theta}}$ is parameterized by an ICNN and try to address the following problem:

***Problem:*** *How to solve (P) efficiently in the setting of ICNN?*

## 2 CHALLENGES AND MOTIVATIONS

Numerous efforts have been made in the literature to study algorithms for optimizing convex functions, in particular in variational reconstruction. Gradient methods are often applied to general smooth convex problems (Boyd & Vandenberghe, 2004) and can be extended to subgradient methods for non-smooth problems (Boyd et al., 2003). Another essential component for non-smooth problems is the proximal operator (Parikh & Boyd, 2014). In particular, primal-dual methods have been extensively studied for non-smooth handcrafted regularizers such as TV (Chambolle & Pock, 2011; 2016; Yan, 2018; Zhu & Chan, 2008). However, due to the nested structure of neural networks, computing the proximal operator for neural networks is often impractical. Therefore, to perfom variational reconstruction with neural network-parameterized regularizers, subgradient methods are commonly applied, where subgradients are computed via backpropagation (Mukherjee et al., 2020). Despite the simplicity of this approach, challenges arise due to non-smoothness.

On the other hand, (Askari et al., 2018; Carreira-Perpinan & Wang, 2014; Li et al., 2019; Taylor et al., 2016; Wang & Benning, 2023; Zhang & Brand, 2017) explored unconventional training methods of training neural network. They proposed to remove nested structure of the neural network by introducing auxiliary variables given by the layer-wise activations. Relaxed problems are considered by introducing penalty to the induced equality constraints. However, the problem remains non-convex, and the minimizers are altered as a result of these relaxations.

## 2.1 CONTRIBUTIONS

Primal-dual algorithms have been successfully applied to classical variational problems, providing fast reconstruction methods. Motivated by their flexibility and practicality, we aim to exploit both the inherent convex nature and the architecture of the neural network to devise optimization algorithm for solving the variational problem. Our contributions are as follows:

- We introduce a more general architecture than ICNN. To address the non-smoothness and nested structure, we propose a novel reformulation of the variational problem. We prove that this reformulation is both convex and equivalent to the original variational problem.

- We apply this novel convex reformulation to setting where the regularizer is parameterized by an ICNN, solving the associated variational problem using a primal-dual algorithm. Additionally, we design a step-size scheme tailored specifically to our formulation.

- We implement the proposed framework for image restoration tasks such as denoising, in-painting, and CT reconstruction. Our results demonstrate that the proposed method is superior to subgradient methods, achieving faster and more stable reconstruction.

# 3 PROPOSED METHOD

## 3.1 CONSTRAINED CONVEX REFORMULATION

Instead of focusing on the specific ICNN architecture considered before, we present our proposed reformulation in the setting of a more general nested structure for the functional $R_{\boldsymbol{\theta}}$:

$$
\begin{aligned}
\mathbf{z}_1 &= \phi_1(\mathbf{x}), \\
\mathbf{z}_{i+1} &= \phi_{i+1}(\mathbf{x}, \boldsymbol{\omega}_i) \text{ for } i = 1, \dots, L-2, \text{ with } \boldsymbol{\omega}_i = (\mathbf{z}_1, \dots, \mathbf{z}_i), \\
R_{\boldsymbol{\theta}}(\mathbf{x}) &= \phi_L(\mathbf{x}, \boldsymbol{\omega}_{L-1}).
\end{aligned}
\tag{EQ-G}
$$

We make the following assumption on the activation functions.

**Assumption 1.** $\phi_i$ are convex for $i = 1, \dots, L$, and $\phi_i^{\mathbf{x}}$ are non-decreasing for $i = 2, \dots, L$, where $\phi_i^{\mathbf{x}}(\boldsymbol{\omega}_{i-1}) = \phi_i(\mathbf{x}, \boldsymbol{\omega}_{i-1})$.

**Proposition 1.** *Under Assumption 1, $R_{\boldsymbol{\theta}}$ defined by (EQ-G) is convex with respect to $\mathbf{x}$.*

Note that the ICNN architecture given by (EQ) is a special case of the above structure, with $\phi_{i+1}(\mathbf{x}, \boldsymbol{\omega}_i) = h_{i+1}(\mathbf{V}_i \mathbf{x} + \mathbf{W}_i \mathbf{z}_i + \mathbf{b}_i)$. In particular, $\mathbf{W}_i$ being non-negative and $h_{i+1}$ being non-decreasing imply that $\phi_{i+1}^{\mathbf{x}}$ is non-decreasing. Hence, $R_{\boldsymbol{\theta}}$ parametrized as in (EQ) is indeed convex. We also relax the condition on $h_1$ to be merely convex, rather than both convex and non-decreasing, as in Amos et al. (2017). With the above framework, we could also consider a residual architecture, where $\phi_{i+1}(\mathbf{x}, \boldsymbol{\omega}_i) = \mathbf{z}_i + h_{i+1}(\mathbf{V}_i \mathbf{x} + \mathbf{W}_i \mathbf{z}_i + \mathbf{b}_i)$. The proofs of Proposition 1 and all following results are deferred to the Appendix.

The main objective of this paper is to minimize a functional $R_{\boldsymbol{\theta}}$ with the above structure efficiently. The first step of the proposed approach involves removing the nested structure of the problem. Given $R_{\boldsymbol{\theta}}$ as defined in (EQ-G), the problem (P) is equivalent to Carreira-Perpinan & Wang (2014):

$$
\min_{\mathbf{x}, \boldsymbol{\omega}_{L-1}} D(\mathbf{A}\mathbf{x}, \mathbf{y}) + \gamma \phi_L(\mathbf{x}, \boldsymbol{\omega}_{L-1}) \text{ subject to } \boldsymbol{\omega}_{L-1} \text{ satisfying (EQ-G).}
\tag{1}
$$

However, the above reformulation is in general not convex as $\phi_i$ could be non-linear.

**Example.** To illustrate the non-convexity of (1), consider a simple 1D example. Here, we define $R_{\boldsymbol{\theta}}(x) = \exp(x + \max(x, 0))$ and a data fidelity $D(x, y) = \frac{1}{2}(x - y)^2$. Then reformulation (1) can be written as:

$$
\min_{x, z} \frac{1}{2}(x - y)^2 + \exp(x + z) \text{ subject to } z = \max(x, 0).
$$

Here $\mathbf{w}_1 = (-1, 0)$, $\mathbf{w}_2 = (1, 1)$ are both feasible but $0.5\mathbf{w}_1 + 0.5\mathbf{w}_2 = (0, 0.5)$ is not. Hence, the feasible set of the above problem is non-convex, so the above problem is non-convex despite that the objective is convex. This is due to the fact that the graph (red curve) of the $\max$ function is not convex. However, $0.5\mathbf{w}_1 + 0.5\mathbf{w}_2$ belongs to the shaded region given by $\{(x, z) | z \geq \max(x, 0)\}$, which is the epigraph of $\max$. In fact, epigraphs can represent a large family of non-linear constraints which are effective in inverse problems. Epigraphical projections were applied in (Chierchia et al., 2015) to solve classes of constrained convex optimization problems.

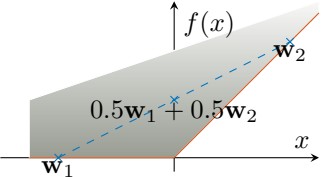

Figure 1: Example illustrating the non-convexity of (1).

This motivates modifying the constraints in (EQ-G) as:

$$
\begin{aligned}
\mathbf{z}_1 &\geq \phi_1(\mathbf{x}), \\
\mathbf{z}_{i+1} &\geq \phi_{i+1}(\mathbf{x}, \boldsymbol{\omega}_i), \ i = 1, \dots, L-2.
\end{aligned}
\tag{IQ-G}
$$

**Proposition 2.** *Given* $\mathbf{x}$*, we define the sets* $E(\mathbf{x}) := \{\boldsymbol{\omega}_{L-1}|\boldsymbol{\omega}_{L-1} \text{ satisfies } (EQ - G)\}, I(x) := \{\boldsymbol{\omega}_{L-1}|\boldsymbol{\omega}_{L-1} \text{ satisfies } (IQ - G)\}$. *Under Assumption 1,* $R_{\boldsymbol{\theta}}$ *defined by (EQ-G) satisfies*

$$R_{\boldsymbol{\theta}}(\mathbf{x}) = \min_{\boldsymbol{\omega}_{L-1} \in E(\mathbf{x})} \phi_L(\mathbf{x}, \boldsymbol{\omega}_{L-1}) = \min_{\boldsymbol{\omega}_{L-1} \in I(\mathbf{x})} \phi_L(\mathbf{x}, \boldsymbol{\omega}_{L-1}). \tag{2}$$

We make the following assumption on the data fidelity and the regularization parameter.

**Assumption 2.** $D(\mathbf{Ax}, \mathbf{y})$ *is convex in* $\mathbf{x}$ *and* $\gamma > 0$.

**Theorem 3.** *Under Assumptions 1 and 2, the following problem is convex*

$$\min_{x, \boldsymbol{\omega}_{L-1}} D(\mathbf{Ax}, \mathbf{y}) + \gamma \phi_L(\mathbf{x}, \boldsymbol{\omega}_{L-1}) \text{ subject to } \boldsymbol{\omega}_{L-1} \text{ satisfies (IQ-G)}. \tag{3}$$

*Furthermore, we denote* $\mathcal{S}_1$ *as set of minimizers of (P) with* $R_{\boldsymbol{\theta}}$ *defined by (EQ-G), and* $\mathcal{S}_2$ *as set of minimizers of (3). Then* $\hat{\mathbf{x}} \in \mathcal{S}_1$ *if and only if there exists* $\hat{\boldsymbol{\omega}}_{L-1}$ *such that* $(\hat{\mathbf{x}}, \tilde{\boldsymbol{\omega}}_{L-1}) \in \mathcal{S}_2$.

**Corollary 4.** *Consider the problem (P) with* $R_{\boldsymbol{\theta}}$ *given by an ICNN. Under Assumption 2, the following problem is convex*

$$\min_{\mathbf{x}, \mathbf{z}_1, \dots, \mathbf{z}_{L-1}} D(\mathbf{Ax}, \mathbf{y}) + \gamma h_L(\mathbf{V}_{L-1}\mathbf{x} + \mathbf{W}_{L-1}\mathbf{z}_{L-1} + \mathbf{b}_{L-1})$$

$$\text{subject to } \mathbf{z}_1 \geq h_1(\mathbf{V}_0\mathbf{x} + \mathbf{b}_0), \tag{P1}$$

$$\mathbf{z}_{i+1} \geq h_{i+1}(\mathbf{V}_i\mathbf{x} + \mathbf{W}_i\mathbf{z}_i + \mathbf{b}_i), \; i = 1, \dots, L - 2.$$

*Furthermore,* $\hat{\mathbf{x}}$ *is a minimizer of (P) if and only if there exists* $\hat{\mathbf{z}}_1, \dots, \hat{\mathbf{z}}_{L-1}$ *such that* $(\hat{\mathbf{x}}, \hat{\mathbf{z}}_1, \dots, \hat{\mathbf{z}}_{L-1})$ *is a minimizer of (P1).*

### 3.2 PRIMAL-DUAL FRAMEWORK

The final step of the proposed framework for solving (P1) is to replace the inequality constraints by indicator functions and reformulate (P1) as an equivalent unconstrained problem:

$$\min_{\mathbf{x}, \mathbf{z}_1, \dots, \mathbf{z}_{L-1}} D(\mathbf{Ax}, \mathbf{y}) + \gamma h_L(\mathbf{V}_{L-1}\mathbf{x} + \mathbf{W}_{L-1}\mathbf{z}_{L-1} + \mathbf{b}_{L-1})$$

$$+ \delta_{C_1}(\mathbf{V}_0\mathbf{x} + \mathbf{b}_0, \mathbf{z}_1) + \sum_{i=2}^{L-1} \delta_{C_i}(\mathbf{V}_{i-1}\mathbf{x} + \mathbf{W}_{i-1}\mathbf{z}_{i-1} + \mathbf{b}_{i-1}, \mathbf{z}_i), \tag{4}$$

here $C_i := \{(p, q)|h_i(p) \leq q\}$, and the indicator function is given by $\delta_{C_i}(\mathbf{x})$ which is 0 if $(p, q) \in C_i$ and $\infty$ otherwise. We then apply a primal-dual algorithm to solve (4).

To utilize the PDHG algorithm Chambolle & Pock (2011); Esser et al. (2010); Zhu & Chan (2008), we recast (4) in the following form:

$$\min_{\mathbf{u}} \left\{ \sum_{i=0}^L f_i(\mathbf{K}_i\mathbf{u}) + g(\mathbf{u}) \right\}. \tag{5}$$

We introduce the variable $\mathbf{u} = (\mathbf{x}, \mathbf{z}_1, \dots, \mathbf{z}_{L-1})$ and consider:

$$\mathbf{K}_1 = \begin{pmatrix} \mathbf{V}_0 & \mathbf{0} & \mathbf{0} & \cdots & \mathbf{0} \\ \mathbf{0} & \mathbf{I} & \mathbf{0} & \cdots & \mathbf{0} \end{pmatrix}$$

$$\mathbf{K}_i = \begin{pmatrix} \mathbf{V}_{i-1} & \mathbf{0} & \cdots & \mathbf{0} & \mathbf{W}_{i-1} & \mathbf{0} & \cdots & \mathbf{0} & \mathbf{0} \\ \mathbf{0} & \mathbf{0} & \cdots & \mathbf{0} & \mathbf{0} & \mathbf{I} & \mathbf{0} & \cdots & \mathbf{0} \end{pmatrix}, i = 2, \dots, L - 1 \tag{6}$$

$$\mathbf{K}_L = \begin{pmatrix} \mathbf{V}_{L-1} & \mathbf{0} & \cdots & \mathbf{0} & \mathbf{0} & \mathbf{0} & \cdots & \mathbf{0} & \mathbf{W}_{L-1} \end{pmatrix}$$

$$\boldsymbol{\beta}_i = \begin{pmatrix} \mathbf{b}_{i-1} \\ \mathbf{0} \end{pmatrix}, i = 1, \dots, L.$$

The data fidelity term $D(\mathbf{Ax}, \mathbf{y})$ can either be included as $f_0(\mathbf{K}_0\mathbf{u})$ or as $g(\mathbf{u})$, where $\mathbf{K}_0 = \begin{pmatrix} \mathbf{A} & \mathbf{0} & \cdots & \mathbf{0} \end{pmatrix}$.

We then consider the following updates of PDHG: We then consider the following updates of PDHG:

$$\mathbf{u}^{k+1} = \text{prox}_g^{\mathbf{T}}(\mathbf{u}^k - \mathbf{TK}^*\mathbf{v}^k)$$

$$\bar{\mathbf{u}}^{k+1} = \mathbf{u}^{k+1} + \theta\left(\mathbf{u}^{k+1} - \mathbf{u}^k\right) \tag{7}$$

$$\mathbf{v}_i^{k+1} = \text{prox}_{f_i^*}^{\mathbf{S}_i}(\mathbf{v}_i^k + \mathbf{S}_i\mathbf{K}_i\bar{\mathbf{u}}^{k+1}), \; i = 0, \dots, L,$$

here the proximal operators are defined as $\text{prox}_h^{\mathbf{S}}(\mathbf{x}) = \arg\min_{\mathbf{x}'} \left\{ \frac{1}{2}\|\mathbf{x}' - \mathbf{x}\|_{\mathbf{S}^{-1}}^2 + h(\mathbf{x}) \right\}$, where $\|\mathbf{x}\|_{\mathbf{S}^{-1}}^2 = \langle \mathbf{x}, \mathbf{S}^{-1}\mathbf{x} \rangle$, and the step-size matrices $\mathbf{T}, \mathbf{S}_i$ are symmetric and positive definite. The algorithm is known to converge Pock & Chambolle (2011) if $\|\mathbf{S}^{1/2}\mathbf{K}\mathbf{T}^{1/2}\| < 1$ and $\theta = 1$, where $\mathbf{S} = \text{diag}(\mathbf{S}_1, \ldots, \mathbf{S}_n)$. We choose diagonal matrices $\mathbf{T}, \mathbf{S}_i$ as our step-size matrices. Applying the Moreau identity, which relates the proximal operator of a function $h$ to that of its conjugate $h^*$ defined by $h^*(\mathbf{y}) = \sup_{\mathbf{x}} \langle \mathbf{x}, \mathbf{y} \rangle - h(\mathbf{x})$, updates for $\mathbf{v}_i$s can be computed via $\text{prox}_{f_i}$, which are the projections onto $C_i$ or $\text{prox}_{h_L}$. With common choices of activations such as ReLU, leaky ReLU, these operators can be computed exactly. More details can be found in the Appendix.

The proposed primal-dual framework introduces auxiliary variables $\mathbf{z}_i$. However, these auxiliary variables correspond directly to the layer-wise activations already present in the network. Hence, the method does not incur additional memory costs compared to standard backpropagation (Li et al., 2019). Moreover, the updates for the auxiliary variables and the dual variables can be computed independently, which offers the potential for efficient parallel computation.

## 4 EXPERIMENTS

We evaluate the performance of the proposed method and compare with subgradient methods on three imaging tasks, (i) salt and pepper denoising, (ii) image inpainting, and (iii) sparse-view CT reconstruction. For all tasks, we utilize a learned regularizer parametrized by an ICNN, which consists of a convolution layer and a global average operator layer, followed by two fully connected layers. The regularizer can be represented by $R_{\boldsymbol{\theta}}(\mathbf{x}) = \mathbf{W}_2 h_2(\mathbf{W}_1 \mathbf{P}\mathbf{z} + \mathbf{b}_1)$ with $\mathbf{z} = h_1(\mathbf{V}_0 x + \mathbf{b}_0)$. Here $\mathbf{V}_0$ corresponds to a convolution operator with 32 $5 \times 5$ filters, and $\mathbf{P}$ denotes an average pooling operater with $16 \times 16$ pool size. The fully connected layers $\mathbf{W}_1, \mathbf{W}_2$ consists of 256 and 1 output neurons respectively. The activations $h_1, h_2$ are chosen to be leaky ReLU and ReLU respectively, with the leaky ReLU's negative slope set to 0.2. The regularizer is then trained following the adversarial framework Lunz et al. (2018), taking (possibly un-paired) ground truth signals as positive samples and unregularized reconstructions, which are task-dependent, as negative samples. The associated minimization problem is then solved with the proposed method, and compare with the subgradient method (Boyd et al., 2003) with (a) constant step-size (SM-C) and (b) diminishing step-size (SM-D), with step-size at the $k$-th iteration given by the initial step-size divided by $k$. More details on adversarial training and the subgradient methods can be found in the Appendix.

### 4.1 SALT AND PEPPER DENOISING

In this example, 1000 grayscale images from the FFHQ dataset (Karras et al., 2019) downsampled to size $256 \times 256$ are used as training data. The salt and pepper corrupted images are used as negative samples in adversarial training. To deal with salt and pepper noise, we utilize an $L^1$-data fidelity (Chambolle & Pock, 2011). The optimization problem is formulated as:

$$\min_{\mathbf{x}, \mathbf{z}} \lambda\|\mathbf{x} - \mathbf{y}\|_1 + \mathbf{W}_2 h_2(\mathbf{W}_1 \mathbf{P}\mathbf{z} + \mathbf{b}_1) + \delta_{C_1}(\mathbf{V}_0\mathbf{x} + \mathbf{b}_0, \mathbf{z}), \tag{8}$$

where $C_1 = \{(p, q)|h_1(p) \leq q\}$. The steps to solve the variational problem are outlined as follows:

$$\mathbf{x}^{k+1}, \mathbf{z}^{k+1} = \text{prox}_{\lambda\|\cdot - \mathbf{y}\|_1}^{\tau_1}(\mathbf{x}^k - \tau_1 \mathbf{V}_0^* \mathbf{v}_{1,1}^k), \mathbf{z}^k - \tau_2(\mathbf{v}_{1,2}^k + \mathbf{P}^* \mathbf{W}_1^* \mathbf{v}_2^k)$$

$$\bar{\mathbf{x}}^{k+1}, \bar{\mathbf{z}}^{k+1} = 2\mathbf{x}^{k+1} - \mathbf{x}^k, 2\mathbf{z}^{k+1} - \mathbf{z}^k$$

$$(\tilde{\mathbf{v}}_{1,1}^{k+1}, \tilde{\mathbf{v}}_{1,2}^{k+1}), \tilde{\mathbf{v}}_2^{k+1} = (\mathbf{v}_{1,1}^k + \sigma_1 \mathbf{V}_0\bar{\mathbf{x}}^{k+1}, \mathbf{v}_{1,2}^k + \sigma_1\bar{\mathbf{z}}^{k+1}), \mathbf{v}_2^k + \sigma_2 \mathbf{W}_1\mathbf{P}\bar{\mathbf{z}}^{k+1} \tag{9}$$

$$\mathbf{v}_1^{k+1}, \mathbf{v}_2^{k+1} = \tilde{\mathbf{v}}_1^{k+1} - \sigma\text{proj}_{C_1}\left(\frac{\tilde{\mathbf{v}}_1^{k+1}}{\sigma} + \boldsymbol{\beta}_1\right), \tilde{\mathbf{v}}_2^{k+1} - \text{prox}_{f_2}^{\sigma_2^{-1}}\left(\frac{\tilde{\mathbf{v}}_2^{k+1}}{\sigma_2}\right).$$

We consider vector-valued step-sizes, $\mathbf{T} = \text{diag}(\tau_1 \mathbf{I}_{\mathbf{x}}, \tau_2 \mathbf{I}_{\mathbf{z}}), \mathbf{S} = \text{diag}(\sigma_1 \mathbf{I}_{\mathbf{v}_1}, \sigma_2 \mathbf{I}_{\mathbf{v}_2})$. The step-sizes are chosen based on the condition $\|\mathbf{S}^{1/2}\mathbf{K}\mathbf{T}^{1/2}\| < 1$ and are given by:

$$\sigma_1 = \frac{c_1}{\|\mathbf{V}_0\|^2}, \sigma_2 = \frac{c_2}{\|\mathbf{W}_1\mathbf{P}\|^2}, \tau_1 = \frac{1}{\sigma_1\|\mathbf{V}_0\|^2}, \tau_2 = \frac{1}{\sigma_1 + \sigma_2\|\mathbf{W}_1\mathbf{P}\|^2}, \tag{10}$$

with hyperparameters $c_1, c_2$. Details on the step-size selection scheme can be found in the Appendix.

**Parameters:** For this experiment, we set the gradient penalty for adversarial training as 5 and set $\lambda = 0.02$. For the proposed method, we pick $c_1, c_2$ from $\{5e\text{-}3, 1e\text{-}2, 5e\text{-}2, 1e\text{-}1, 5e\text{-}1, 1, 5\}$, $\{5e\text{-}6, 1e\text{-}5, 5e\text{-}5, 1e\text{-}4, 5e\text{-}4, 1e\text{-}3, 5e\text{-}3\}$. For SM-C, we choose the step-size from $\{0.1, 0.5, 1, 2\}$. As for SM-D, we select the initial step-size from $\{1, 3, 5, 10\}$.

**Ablation study:** Figure 2 shows the ablation study of the step-size hyperparameters for the proposed method. We ran 200 iterations of the proposed method for each hyperparameter combination and evaluated the average objective value to assess convergence. The left plot shows the average objective values, while the right plot depicts energy versus iterations for different values of $c_1, c_2$.

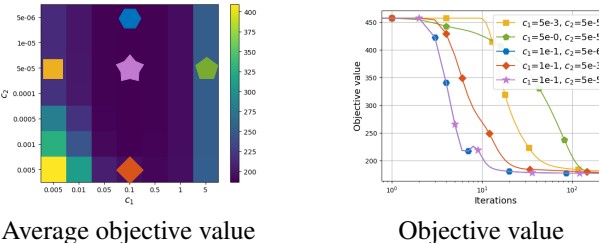

Average objective value            Objective value

Figure 2: Denoising: Ablation study of proposed method for step-size hyperparameters. The markers on the left corresponds to those depicted in the energy versus iterations plots on the right.

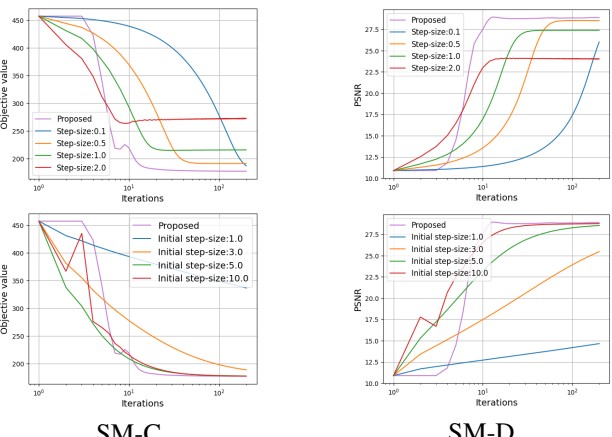

SM-C                    SM-D

Figure 3: Denoising: Comparison to subgradient methods.

**Results:** The proposed method with the optimal choice of $c_1, c_2$ ($c_1, c_2 = 1e\text{-}1, 5e\text{-}5$) is then compared with the subgradient methods. The first row of Figure 3 shows energy versus iterations plots, indicating that SM-C fail to converge within 200 iterations, while SM-D do converge, albeit slower than the proposed method. Moreover, we evaluate the Peak Signal-to-Noise Ratio (PSNR). The proposed method achieves the highest PSNR values in less than 20 iterations, outperforming both subgradient methods. Figure 4 shows the reconstructed images produced by each method. Reconstructions are provided at both 15 and 200 iterations, with the proposed method delivering visually satisfactory results as early as 15 iterations.

## 4.2 IMAGE INPAINTING

We consider an image inpainting task in this section. We randomly remove 30% of the pixels of the image. We further add Gaussian noise with standard deviation of 0.03 to the masked image. Given the noise model, we adopt a $L^2$ data term and formulate the optimization problem as:

$$\min_{\mathbf{x}, \mathbf{z}} \frac{1}{2}\|\mathbf{A}\mathbf{x} - \mathbf{y}\|_2^2 + \gamma \mathbf{W}_2 h_2(\mathbf{W}_1 \mathbf{P}\mathbf{z} + \mathbf{b}_1) + \delta_{C_1}(\mathbf{V}_0 \mathbf{x} + \mathbf{b}_0, \mathbf{z}), \tag{11}$$

where $\mathbf{A}$ is a binary diagonal matrix that corresponds to the sampling mask. The regularizer is trained with the same dataset in the previous experiment with the masked noisy images as negative

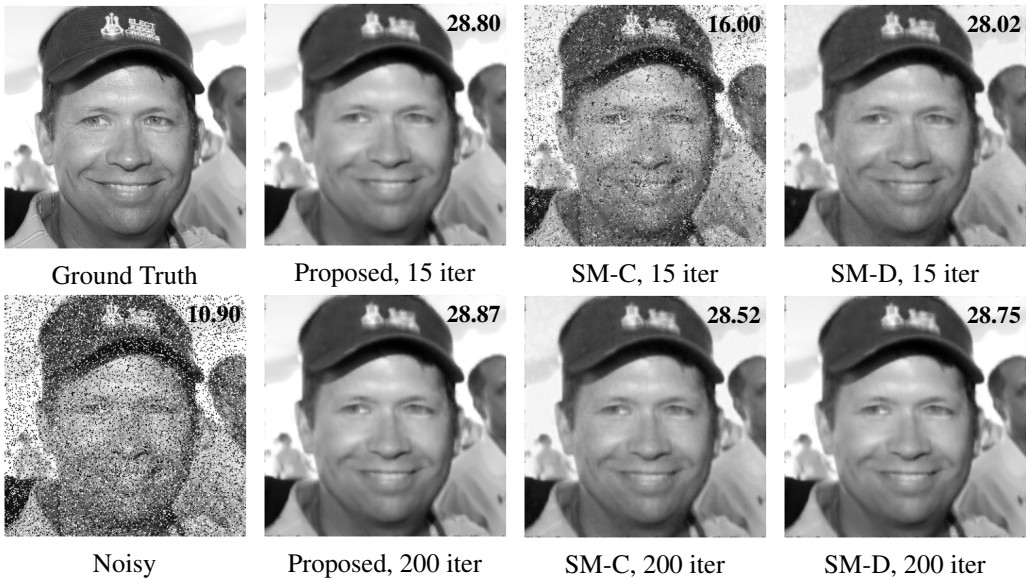

Figure 4: Denoising: Visual comparison of reconstructions, with PSNR values shown in the top right corner. The proposed method ($c_1, c_2$=1e-1, 5e-5) achieves a visually satisfactory reconstruction within 15 iterations, while that of SM-C remains noisy.

samples. The updates of the primal-dual framework are as in (9), with the $L^1$ data term replaced by the $L^2$ data term. The step-sizes are also chosen following (10).

**Parameters:** We set the gradient penalty for adversarial training as 5 and set $\gamma = 0.1$. We choose $c_1, c_2$ from $\{1e\text{-}4, 5e\text{-}4, 1e\text{-}3, 5e\text{-}3, 1e\text{-}2, 5e\text{-}2, 1e\text{-}1\}$, $\{1e\text{-}6, 5e\text{-}6, 1e\text{-}5, 5e\text{-}5, 1e\text{-}4, 5e\text{-}4, 1e\text{-}3\}$. For SM-C, we select the step-sizes from $\{0.5, 1, 1.5, 2\}$. For SM-D, the initial step-sizes are chosen from $\{10, 30, 50, 60\}$.

Table 1: Comparisons to subgradient methods across dataset.

| Methods | Iterations (Mean$\pm$Std) | Speedup |
|---------|---------------------------|---------|
| Proposed | 85.5$\pm$9.31 | — |
| SM-C | 151.6$\pm$23.50 | 1.81 |
| SM-D | 266.6$\pm$18.67 | 3.17 |

**Results:** To evaluate the performance of the proposed method across different test images, we solved the minimization problem on 20 test images and recorded the number of iterations required to reduce the relative objective error below $1e\text{-}3$. Table 1 shows the mean and standard deviation of the number of iterations needed for all methods. Additionally, the mean speedup of the proposed method compared to the subgradient methods is reported, demonstrating the efficiency of the proposed method, highlighting its efficiency. Figure 5 show the comparisons of energy and PSNR plots. Notably, the proposed method converges significantly faster compared to both subgradient approaches. Interestingly, unlike the previous experiment, the diminishing step-size does not improve the convergence speed. It is important to note that while the proposed method effectively minimizes the objective function, it does not yield the reconstruction with the best PSNR. This discrepancy may be due to the fact that the regularizer was trained using an unsupervised adversarial framework. Figure 6 shows the reconstructions, illustrating that despite the PSNR drop, the proposed method still produces visually appealing reconstructions in considerably fewer iterations than the subgradient methods.

### 4.3 CT WITH POISSON NOISE

In this section, we consider a sparse-view computed tomography (CT) reconstruction task, with human abdominal CT scans of the Mayo clinic for the low-dose CT grand challenge (McCollough,

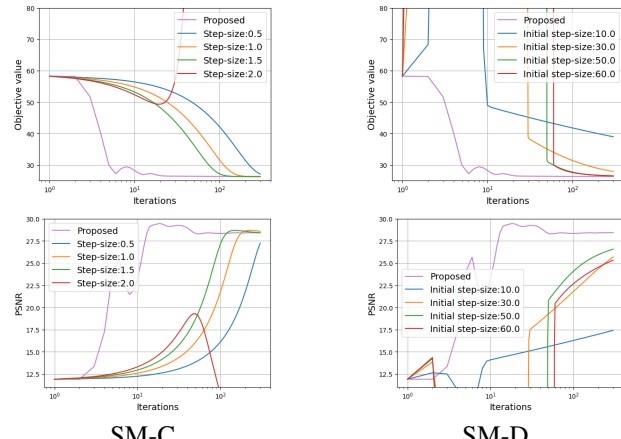

SM-C  SM-D

Figure 5: Inpainting: Comparison to subgradient methods ($c_1, c_2$=5e-3, 5e-5).

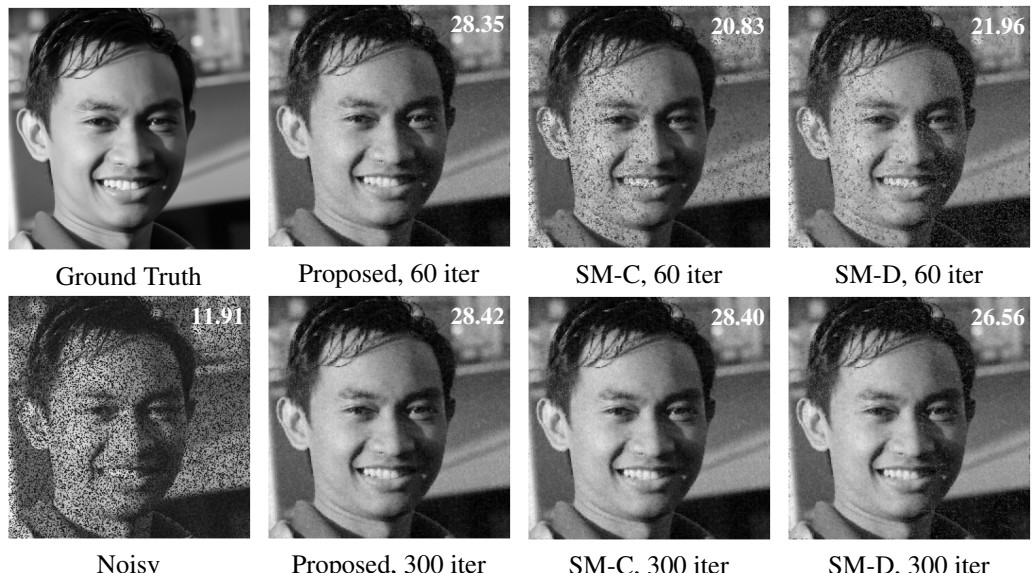

| Ground Truth | Proposed, 60 iter | SM-C, 60 iter | SM-D, 60 iter |
| Noisy | Proposed, 300 iter | SM-C, 300 iter | SM-D, 300 iter |

Figure 6: Inpainting: Visual comparison of reconstructions, with PSNR shown at top right corner.

2016) as training and testing data. The measurements are simulated using a parallel beam geometry with 200 angles and 400 bins. We model the noise as Poisson with a constant background level $r = 50$. The filtered back projections (FBP) are used as negative samples for training. We also assumed that the reconstruction is bounded below by 0. We consider the Kullback–Leibler (KL) divergence data fidelity, which is suitable for Poisson-distributed data:

$$D(\mathbf{Ax}, \mathbf{y}) = \mathbf{1}^T (\mathbf{Ax} - \mathbf{y} + \mathbf{r}) + \mathbf{y}^T \log \left( \frac{\mathbf{y}}{\mathbf{Ax} + \mathbf{r}} \right), \tag{12}$$

where $\mathbf{1}$ denotes a vector of 1s. We consider the following optimization problem:

$$\min_{\mathbf{x}, \mathbf{z}} D(\mathbf{Ax}, \mathbf{y}) + \gamma \mathbf{W}_2 h_2(\mathbf{W}_1 \mathbf{Pz} + \mathbf{b}_1) + \delta_{C_1}(\mathbf{V}_0 \mathbf{x} + \mathbf{b}_0, \mathbf{z}) + \delta_{[0, \infty)}(\mathbf{x}), \tag{13}$$

where $\mathbf{A}$ is the scaled X-ray transform with the prescribed geometry.

Unlike the previous experiment, we dualize the forward operater $\mathbf{A}$ with the data fidelity acting as $f_0$. This leads to the following updates:

$$\mathbf{x}^{k+1}, \mathbf{z}^{k+1} = \max(\mathbf{x}^k - \tau_1 \mathbf{A}^* \mathbf{v}_0 + \mathbf{V}_0^* \mathbf{v}_{1,1}^k, \mathbf{0}), \mathbf{z}^k - \tau_2(\mathbf{v}_{1,2}^k + \mathbf{P}^* \mathbf{W}_1^* \mathbf{v}_2^k)$$

$$\overline{\mathbf{x}}^{k+1}, \overline{\mathbf{z}}^{k+1} = 2\mathbf{x}^{k+1} - \mathbf{x}^k, 2\mathbf{z}^{k+1} - \mathbf{z}^k$$

$$(\tilde{\mathbf{v}}_{1,1}^{k+1}, \tilde{\mathbf{v}}_{1,2}^{k+1}), \tilde{\mathbf{v}}_2^{k+1} = (\mathbf{v}_{1,1}^k + \sigma_1 \mathbf{V}_0 \overline{\mathbf{x}}^{k+1}, \mathbf{v}_{1,2}^k + \sigma_1 \overline{\mathbf{z}}^{k+1}), \mathbf{v}_2^k + \sigma_2 \mathbf{W}_1 \mathbf{P} \overline{\mathbf{z}}^{k+1}$$

$$\mathbf{v}_0^{k+1} = \mathrm{prox}_{f_0^*}^{\sigma_0}(\mathbf{v}_0^k + \sigma_0 \mathbf{A} \overline{\mathbf{x}}^{k+1})$$

$$\mathbf{v}_1^{k+1}, \mathbf{v}_2^{k+1} = \tilde{\mathbf{v}}_1^{k+1} - \sigma \mathrm{proj}_{C_1}\left(\frac{\tilde{\mathbf{v}}_1^{k+1}}{\sigma} + \boldsymbol{\beta}_1\right), \tilde{\mathbf{v}}_2^{k+1} - \mathrm{prox}_{f_2}^{\sigma_2^{-1}}\left(\frac{\tilde{\mathbf{v}}_2^{k+1}}{\sigma_2}\right). \tag{14}$$

Similarly, we pick step-sizes $\mathbf{T} = \mathrm{diag}(\tau_1 \mathbf{I_x}, \tau_2 \mathbf{I_z})$, $S = \mathrm{diag}(\sigma_0 \mathbf{I_{v_0}}, \sigma_1 \mathbf{I_{v_1}}, \sigma_2 \mathbf{I_{v_2}})$ given by:

$$\sigma_0 = \frac{c_0}{\|\mathbf{A}\|^2}, \sigma_1 = \frac{c_1}{\|\mathbf{V}_0\|^2}, \sigma_2 = \frac{c_2}{\|\mathbf{W}_1 \mathbf{P}\|^2}, \tau_1 = \frac{1}{\sigma_0 \|\mathbf{A}\|^2 + \sigma_1 \|\mathbf{V}_0\|^2}, \tau_2 = \frac{1}{\sigma_1 + \sigma_2 \|\mathbf{W}_1 \mathbf{P}\|^2}. \tag{15}$$

**Parameters:** We set the gradient penalty for adversarial training as $10$ and set $\gamma = 400$. We pick $c_0, c_1, c_2$ from $\{50, 100, 500, 1000, 5000\}, \{10, 50, 100, 500, 1000\}, \{0.1, 0.5, 1, 5, 10\}$. For SM-C, we select step-sizes from $\{0.002, 0.003, 0.004, 0.0005\}$, and $\{0.001, 0.005, 0.01, 0.05\}$ as initial step-sizes for SM-D.

**Results:** Figure 7 compares the energy and PSNR plots of the proposed method and subgradient methods. While the constant step-size subgradient methods show substantial progress in the early iterations, they are quickly surpassed by the proposed method, which demonstrates a more consistent convergence. In contrast, the diminishing step-size subgradient methods exhibit much slower convergence overall.

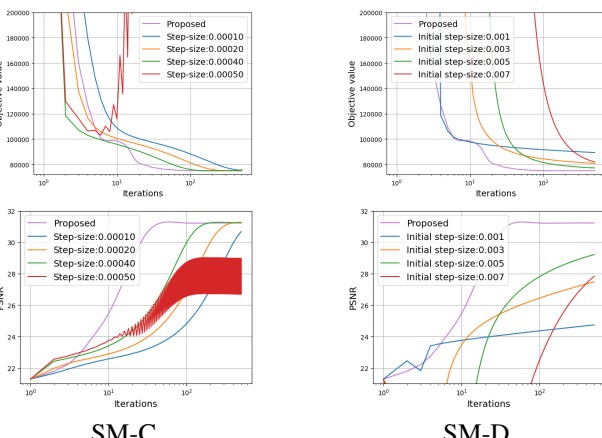

SM-C    SM-D

Figure 7: CT: Comparison to subgradient methods ($c_0, c_1, c_2$=500, 100, 1e-1).

Additionally, Figure 8 shows comparisons of the data fidelity and regularization term plots. All methods handle the data fidelity term smoothly, though the constant step-size subgradient methods can sometimes reach a far lower value than the eventual converged value, which may also explain their initial speed. In contrast, the subgradient methods exhibit different behavior with the non-smooth regularization term. The constant step-size subgradient methods reduce the regularization term much more slowly than the proposed method, while the diminishing step-size subgradient methods show very oscillatory behavior in the initial states, indicating superior stability of the proposed method throughout the optimization process. Figure 9 shows the reconstructions at 50 and 500 iterations, further illustrating the effectiveness of the proposed method in producing high-quality results consistently.

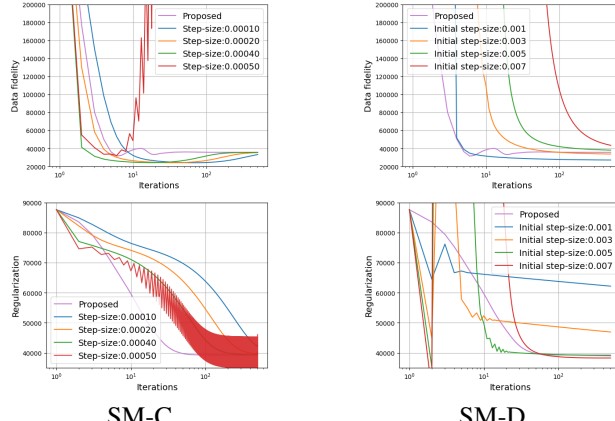

SM-C                                  SM-D

Figure 8: CT: Data fidelity and regularization versus iterations plots. Notably, the subgradient methods with large step sizes exhibit oscillatory behavior, while the proposed method demonstrates more stable convergence.

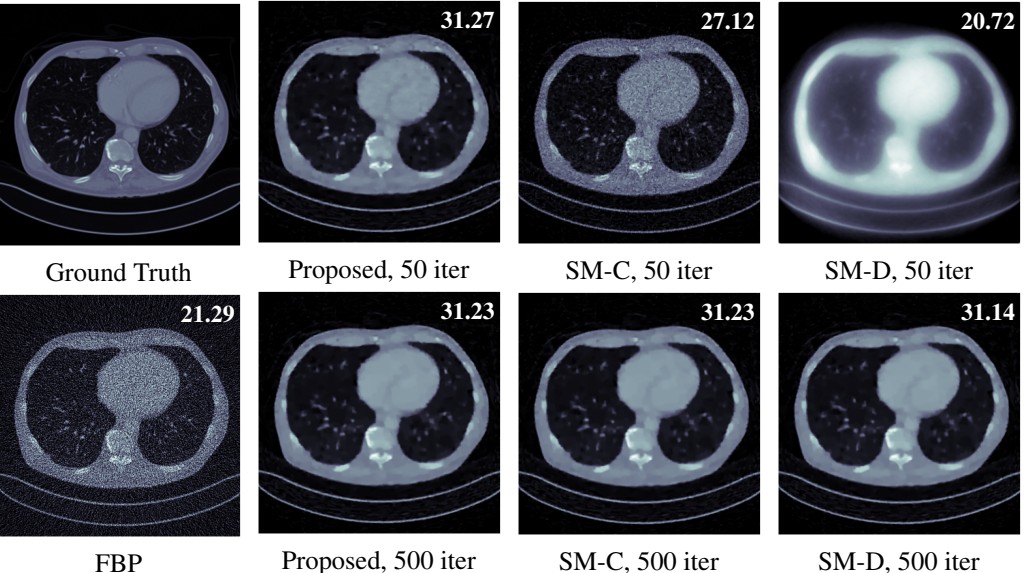

Figure 9: CT: Visual comparison of reconstructions, with PSNR shown at top right corner.

## 5 CONCLUSION

We proposed an efficient method for solving the optimization problem in variational reconstruction with a learned convex regularizer. A key challenge comes from the non-smoothness of the ICNN regularizer, whose proximal operator lacks a closed-form solution. To overcome this, we decoupled the neural network layers by introducing auxiliary variables corresponding to the layer-wise activations. While this initially resulted in a non-convex problem, we drew inspiration from the convexity of epigraphs and reformulated it as a convex optimization problem. We then proved that this reformulation is equivalent to the original variational problem and applied a primal-dual algorithm to solve it. Numerical experiments demonstrated that the proposed method not only outperforms subgradient methods in terms of convergence speed but also exhibits greater stability throughout the optimization process, as evidenced by the smoother behavior observed in the energy versus iterations plots, as well as those depicting data fidelity and regularization. Additionally, we note that the updates of the proposed method are independent, enabling parallel computation. Looking forward, we aim to explore the potential of extending the proposed method to primal-dual variants that leverage this, such as coordinate-descent primal-dual algorithms (Fercoq & Bianchi, 2019).

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

# A PROOFS

We now present the proofs for all the results in Section 3.1.

**Proposition 1.** *Under Assumption 1, $R_{\boldsymbol{\theta}}$ defined by (EQ-G) is convex with respect to $\mathbf{x}$.*

*Proof.* Consider $\bar{\mathbf{x}}, \tilde{\mathbf{x}}$, and $\lambda \in [0,1]$. Then

$$\mathbf{z}_1^{\lambda} := \phi_1(\lambda \bar{\mathbf{x}} + (1-\lambda)\tilde{\mathbf{x}}) \leq \lambda \phi_1(\bar{\mathbf{x}}) + (1-\lambda)\phi_1(\tilde{\mathbf{x}}) =: \lambda \bar{\mathbf{z}}_1 + (1-\lambda)\tilde{\mathbf{z}}_1,$$

where the inequality is due the convexity of $\phi_1$. Since $\phi_2^{\mathbf{x}}$ is non-decreasing, we have:

$$\begin{aligned}
\mathbf{z}_2^{\lambda} := \phi_2(\lambda \bar{\mathbf{x}} + (1-\lambda)\tilde{\mathbf{x}}, \boldsymbol{\omega}_1^{\lambda}) \\
\leq \phi_2(\lambda \bar{\mathbf{x}} + (1-\lambda)\tilde{\mathbf{x}}, \lambda \bar{\boldsymbol{\omega}}_1 + (1-\lambda)\tilde{\boldsymbol{\omega}}_1) \\
\leq \lambda \bar{\mathbf{z}}_2 + (1-\lambda)\tilde{\mathbf{z}}_2,
\end{aligned}$$

where the second inequality follows from the convexity of $\phi_2$. Using similar argument, we have:

$$\boldsymbol{\omega}_i^{\lambda} \leq \lambda \bar{\boldsymbol{\omega}}_i + (1-\lambda)\tilde{\boldsymbol{\omega}}_i, \text{ for } i = 2, \ldots, L-1,$$

where $\boldsymbol{\omega}_i$ are defined as $(\mathbf{z}_1, \ldots, \mathbf{z}_i)$ and $\boldsymbol{\omega}_i^{\lambda} := \phi_i(\lambda \bar{\mathbf{x}} + (1-\lambda)\tilde{\mathbf{x}}, \boldsymbol{\omega}_{i-1}^{\lambda})$. In particular:

$$\begin{aligned}
R_{\boldsymbol{\theta}}(\lambda \bar{\mathbf{x}} + (1-\lambda)\tilde{\mathbf{x}}) = \phi_L(\lambda \bar{\mathbf{x}} + (1-\lambda)\tilde{\mathbf{x}}, \boldsymbol{\omega}_{L-1}^{\lambda}) \\
\leq \phi_L(\lambda \bar{\mathbf{x}} + (1-\lambda)\tilde{\mathbf{x}}, \lambda \bar{\boldsymbol{\omega}}_{L-1} + (1-\lambda)\tilde{\boldsymbol{\omega}}_{L-1}) \\
\leq \lambda \phi_L(\bar{\mathbf{x}}, \bar{\boldsymbol{\omega}}_{L-1}) + (1-\lambda)\phi_L(\tilde{\mathbf{x}}, \tilde{\boldsymbol{\omega}}_{L-1}),
\end{aligned}$$

where the first inequality holds since $\phi_L^{\mathbf{x}}$ is non-decreasing and the second inequality is due to the convexity of $\phi_L$. Hence, $R_{\boldsymbol{\theta}}$ is convex with respect to $\mathbf{x}$. $\qquad\square$

**Proposition 2.** *Given $\mathbf{x}$, we define the sets $E(\mathbf{x}) := \{\boldsymbol{\omega}_{L-1} | \boldsymbol{\omega}_{L-1} \text{ satisfies } (EQ-G)\}, I(x) := \{\boldsymbol{\omega}_{L-1} | \boldsymbol{\omega}_{L-1} \text{ satisfies } (IQ-G)\}$. Under Assumption 1, $R_{\boldsymbol{\theta}}$ defined by (EQ-G) satisfies*

$$R_{\boldsymbol{\theta}}(\mathbf{x}) = \min_{\boldsymbol{\omega}_{L-1} \in E(\mathbf{x})} \phi_L(\mathbf{x}, \boldsymbol{\omega}_{L-1}) = \min_{\boldsymbol{\omega}_{L-1} \in I(\mathbf{x})} \phi_L(\mathbf{x}, \boldsymbol{\omega}_{L-1}). \quad (2)$$

*Proof.* Note that $E(\mathbf{x})$ is a singleton, consists of $\hat{\boldsymbol{\omega}}_{L-1}$ which satisfy (EQ-G) given $\mathbf{x}$. Hence, $R_{\boldsymbol{\theta}}(\mathbf{x}) = \min_{\boldsymbol{\omega}_{L-1} \in E(\mathbf{x})} \phi_L(\mathbf{x}, \boldsymbol{\omega}_{L-1})$. Since $E(\mathbf{x}) \subset I(\mathbf{x})$, we have $\min_{\boldsymbol{\omega}_{L-1} \in E(\mathbf{x})} \phi_L(\mathbf{x}, \boldsymbol{\omega}_{L-1}) \geq \min_{\boldsymbol{\omega}_{L-1} \in I(\mathbf{x})} \phi_L(\mathbf{x}, \boldsymbol{\omega}_{L-1})$.
For $\boldsymbol{\omega}_{L-1} = (\mathbf{z}_1, \ldots, \mathbf{z}_{L-1}) \in I(\mathbf{x})$, we have $\mathbf{z}_1 \geq \phi_1(\mathbf{x}) = \hat{\mathbf{z}}_1$, $\mathbf{z}_i \geq \phi_i(\mathbf{x}, \boldsymbol{\omega}_i) = \hat{\mathbf{z}}_i$ for $i = 2, \ldots, L-1$, where $\hat{\boldsymbol{\omega}}_{L-1} = (\hat{\mathbf{z}}_1, \ldots, \hat{\mathbf{z}}_{L-1})$. Therefore, $\hat{\boldsymbol{\omega}}_{L-1} \leq \boldsymbol{\omega}_{L-1}$ for all $\boldsymbol{\omega}_{L-1} \in I(\mathbf{x})$. Since $\phi_L^{\mathbf{x}}$ is non-decreasing, we have:

$$\phi_L(\mathbf{x}, \hat{\boldsymbol{\omega}}_{L-1}) \leq \phi_L(\mathbf{x}, \boldsymbol{\omega}_{L-1}).$$

Therefore, $\min_{\boldsymbol{\omega}_{L-1} \in E(\mathbf{x})} \phi_L(\mathbf{x}, \boldsymbol{\omega}_{L-1}) \leq \min_{\boldsymbol{\omega}_{L-1} \in I(\mathbf{x})} \phi_L(\mathbf{x}, \boldsymbol{\omega}_{L-1})$. Combining with the other inequality, this shows that $\min_{\boldsymbol{\omega}_{L-1} \in E(\mathbf{x})} \phi_L(\mathbf{x}, \boldsymbol{\omega}_{L-1}) = \min_{\boldsymbol{\omega}_{L-1} \in I(\mathbf{x})} \phi_L(\mathbf{x}, \boldsymbol{\omega}_{L-1})$. $\qquad\square$

**Theorem 3.** *Under Assumptions 1 and 2, the following problem is convex*

$$\min_{\mathbf{x}, \boldsymbol{\omega}_{L-1}} D(\mathbf{A}\mathbf{x}, \mathbf{y}) + \gamma \phi_L(\mathbf{x}, \boldsymbol{\omega}_{L-1}) \text{ subject to } \boldsymbol{\omega}_{L-1} \text{ satisfies } (IQ-G). \quad (3)$$

*Furthermore, we denote $\mathcal{S}_1$ as set of minimizers of (P) with $R_{\boldsymbol{\theta}}$ defined by (EQ-G), and $\mathcal{S}_2$ as set of minimizers of (3). Then $\hat{\mathbf{x}} \in \mathcal{S}_1$ if and only if there exists $\hat{\boldsymbol{\omega}}_{L-1}$ such that $(\hat{\mathbf{x}}, \hat{\boldsymbol{\omega}}_{L-1}) \in \mathcal{S}_2$.*

*Proof.* The constraints (IQ-G) are convex since $\phi_i$ are convex. In particular, $D$ and $\gamma \phi_L$ are convex in $x$, then problem (3) is convex. Due to proposition 2, we have:

$$\begin{aligned}
\min_{\mathbf{x}} D(\mathbf{A}\mathbf{x}, \mathbf{y}) + \gamma R_{\boldsymbol{\theta}}(\mathbf{x}) &= \min_{\mathbf{x}, \boldsymbol{\omega}_{L-1} \in E(\mathbf{x})} D(\mathbf{A}\mathbf{x}, \mathbf{y}) + \gamma \phi(\mathbf{x}, \boldsymbol{\omega}_{L-1}) \\
&= \min_{\mathbf{x}} D(\mathbf{A}\mathbf{x}, \mathbf{y}) + \gamma \min_{\boldsymbol{\omega}_{L-1} \in E(\mathbf{x})} \phi_L(\mathbf{x}, \boldsymbol{\omega}_{L-1}) \\
&= \min_{\mathbf{x}} D(\mathbf{A}\mathbf{x}, \mathbf{y}) + \gamma \min_{\boldsymbol{\omega}_{L-1} \in I(\mathbf{x})} \phi_L(\mathbf{x}, \boldsymbol{\omega}_{L-1}) \\
&= \min_{\mathbf{x}, \boldsymbol{\omega}_{L-1} \in I(\mathbf{x})} D(\mathbf{A}\mathbf{x}, \mathbf{y}) + \gamma \phi(\mathbf{x}, \boldsymbol{\omega}_{L-1}).
\end{aligned}$$

Suppose that $\hat{\mathbf{x}} \in \mathcal{S}_1$. We note that we have $\hat{\boldsymbol{\omega}}_{L-1} \in I(\hat{\mathbf{x}})$ for $\hat{\boldsymbol{\omega}}_{L-1} \in E(\hat{\mathbf{x}})$, which is a singleton. Then we have:

$$D(\mathbf{A}\hat{\mathbf{x}}, \mathbf{y}) + \gamma\phi(\hat{\mathbf{x}}, \hat{\boldsymbol{\omega}}_{L-1}) = \min_{\mathbf{x}} D(\mathbf{A}\mathbf{x}, \mathbf{y}) + \gamma R_{\boldsymbol{\theta}}(\mathbf{x})$$

$$= \min_{\mathbf{x}, \boldsymbol{\omega}_{L-1} \in I(\mathbf{x})} D(\mathbf{A}\mathbf{x}, \mathbf{y}) + \gamma\phi(\mathbf{x}, \boldsymbol{\omega}_{L-1}).$$

This shows that $(\hat{\mathbf{x}}, \hat{\boldsymbol{\omega}}_{L-1})$ is indeed a minimizer of (3). Conversely, suppose $(\hat{\mathbf{x}}, \hat{\boldsymbol{\omega}}_{L-1}) \in \mathcal{S}_2$. Then we have:

$$D(\mathbf{A}\hat{\mathbf{x}}, \mathbf{y}) + \gamma R_{\boldsymbol{\theta}}(\hat{\mathbf{x}}) = D(\mathbf{A}\hat{\mathbf{x}}, \mathbf{y}) + \gamma \min_{\boldsymbol{\omega}_{L-1} \in E(\hat{\mathbf{x}})} \phi_L(\hat{\mathbf{x}}, \boldsymbol{\omega}_{L-1})$$

$$= D(\mathbf{A}\hat{\mathbf{x}}, \mathbf{y}) + \gamma \min_{\boldsymbol{\omega}_{L-1} \in I(\hat{\mathbf{x}})} \phi_L(\hat{\mathbf{x}}, \boldsymbol{\omega}_{L-1})$$

$$= D(\mathbf{A}\hat{\mathbf{x}}, \mathbf{y}) + \gamma\phi(\hat{\mathbf{x}}, \hat{\boldsymbol{\omega}}_{L-1})$$

$$= \min_{\mathbf{x}} D(\mathbf{A}\mathbf{x}, \mathbf{y}) + \gamma R_{\boldsymbol{\theta}}(\mathbf{x}).$$

Therefore, we have $\hat{\mathbf{x}} \in \mathcal{S}_1$ if and only if there exists $\hat{\boldsymbol{\omega}}_{L-1}$ such that $(\hat{\mathbf{x}}, \hat{\boldsymbol{\omega}}_{L-1}) \in \mathcal{S}_2$. $\qquad\square$

**Corollary 4.** *Consider the problem (P) with $R_{\boldsymbol{\theta}}$ given by an ICNN. Under Assumption 2, the following problem is convex*

$$\min_{\mathbf{x}, \mathbf{z}_1, \ldots, \mathbf{z}_{L-1}} D(\mathbf{A}\mathbf{x}, \mathbf{y}) + \gamma h_L(\mathbf{V}_{L-1}\mathbf{x} + \mathbf{W}_{L-1}\mathbf{z}_{L-1} + \mathbf{b}_{L-1})$$

*subject to* $\mathbf{z}_1 \geq h_1(\mathbf{V}_0\mathbf{x} + \mathbf{b}_0),$ $\qquad\qquad$ (P1)

$$\mathbf{z}_{i+1} \geq h_{i+1}(\mathbf{V}_i\mathbf{x} + \mathbf{W}_i\mathbf{z}_i + \mathbf{b}_i), \ i = 1, \ldots, L-2.$$

*Furthermore, $\hat{\mathbf{x}}$ is a minimizer of (P) if and only if there exists $\hat{\mathbf{z}}_1, \ldots, \hat{\mathbf{z}}_{L-1}$ such that $(\hat{\mathbf{x}}, \hat{\mathbf{z}}_1, \ldots, \hat{\mathbf{z}}_{L-1})$ is a minimizer of (P1).*

*Proof.* We note that (EQ) is a special case of (EQ-G) with $\phi_{i+1}(\mathbf{x}, \boldsymbol{\omega}_i) = h_{i+1}(\mathbf{V}_i\mathbf{x} + \mathbf{W}_i\mathbf{z}_i + \mathbf{b}_i)$. Therefore, Assumption 1 is satisfied in the ICNN setting. The result directly follows from Theorem 3 with $(\hat{\mathbf{z}}_1, \ldots, \hat{\mathbf{z}}_{L-1}) =: \hat{\boldsymbol{\omega}}_{L-1}$. $\qquad\square$

# B TRAINING DETAILS

We follow the adversarial training framework in (Lunz et al., 2018; Mukherjee et al., 2020) to train the regularizer $R_{\boldsymbol{\theta}}$. In this approach, the regularizer is trained to output low values when provided with true images and higher values for unregularized reconstructions. To ensure that the regularizer transitions smoothly with respect to the input, we incorporate a gradient penalty term into the training objective. This penalty enforces stability of the learned regularizer. The complete training procedure is detailed in Algorithm 1.

Here $\pi_X$ denotes the true image distribution and $\pi_Y$ denotes the measurement distribution, and $\mathbf{A}^{\dagger}$ denotes the pseudo-inverse of the forward operator.

# C ALGORITHM DETAILS

In this section, we write down the subgradient methods we implemented in the numeric sections. We also provide some additional details on the primal-dual algorithm, specifically the exact formulae for proximal operators and the step-size selection scheme in (10) and (15).

## C.1 SUBGRADIENT METHODS

The subgrdradient method with (a) constant-stepsize (SM-C) and (b) diminishing step-size (SM-D) are given in Algorithm 2 and 3 respectively. For both methods, the subgradients are computed using automatic differentiation.

---

**Algorithm 1** Training the ACR (Mukherjee et al., 2020).

---

1: Input: Gradient penalty $\lambda_{\text{gp}}$, initial value of the network parameters $\boldsymbol{\theta}^{(0)} = \{\mathbf{V}_0, \mathbf{W}_1, \mathbf{W}_2, \mathbf{b}_1, \mathbf{b}_2\}$, mini-batch size $n_b$, parameters $(\eta, \beta_1, \beta_2)$ for the Adam optimizer.
2: **for** $m = 1, 2, \cdots$ (until convergence): **do**
3:     Sample $\mathbf{x}_j \sim \pi_X, \mathbf{y}_j \sim \pi_Y$, and $\epsilon_j \sim$ uniform $[0, 1]$
4:     **for** $1 \leq j \leq n_b$ **do**
5:         Compute $\mathbf{x}_j^{(\epsilon)} = \epsilon_j \mathbf{x}_j + (1 - \epsilon_j) \mathbf{A}^\dagger \mathbf{y}_j$.
6:     Compute the training loss for the $m^{\text{th}}$ mini-batch:

$$\mathcal{L}(\boldsymbol{\theta}) := \frac{1}{n_b} \sum_{j=1}^{n_b} \mathcal{R}_{\boldsymbol{\theta}}(\mathbf{x}_j) - \frac{1}{n_b} \sum_{j=1}^{n_b} \mathcal{R}_{\boldsymbol{\theta}}(\mathbf{A}^\dagger \mathbf{y}_j)$$

$$+ \quad \lambda_{\text{gp}} \cdot \frac{1}{n_b} \sum_{j=1}^{n_b} \left( \left\| \nabla \mathcal{R}_{\boldsymbol{\theta}}\left(\mathbf{x}_j^{(\epsilon)}\right) \right\|_2 - 1 \right)^2.$$

7:     Update $\boldsymbol{\theta}^{(m)} = \text{Adam}_{\eta, \beta_1, \beta_2}\left(\boldsymbol{\theta}^{(m-1)}, \nabla_{\boldsymbol{\theta}}\mathcal{L}\left(\boldsymbol{\theta}^{(m-1)}\right)\right)$.
8:     Zero-clip the negative weights in $\mathbf{W}_1, \mathbf{W}_2$ to preserve convexity.
9: Output: Parameter $\boldsymbol{\theta}$ of the trained ACR.

---

**Algorithm 2** SM-C (Boyd et al., 2003).

---

1: Input: Initialization $x^0$, constant step-size $\eta$, maximum number of iterations $N_{max}$.
2: **for** $k = 0, 1, \ldots, N_{max}$ **do**
3:     Compute subgradient $\mathbf{g}^k \in \partial_{\mathbf{x}}(D(\mathbf{A}\mathbf{x}^k, \mathbf{y}) + \gamma R_{\boldsymbol{\theta}}(\mathbf{x}^k))$.
4:     Update $\mathbf{x}^{k+1} = \mathbf{x}^k - \eta \mathbf{g}^k$.

---

**Algorithm 3** SM-D (Boyd et al., 2003).

---

1: Input: Initialization $x^0$, initial step-size $\eta^0$, maximum number of iterations $N_{max}$.
2: **for** $k = 0, 1, \ldots, N_{max}$ **do**
3:     Compute subgradient $\mathbf{g}^k \in \partial_{\mathbf{x}}(D(\mathbf{A}\mathbf{x}^k, \mathbf{y}) + \gamma R_{\boldsymbol{\theta}}(\mathbf{x}^k))$.
4:     Update $\mathbf{x}^{k+1} = \mathbf{x}^k - \eta^k \mathbf{g}^k$.
5:     Update step-size $\eta^{k+1} = \eta^k / k$.

---

## C.2 PROXIMAL OPERATORS

We first consider $f_1(p, q) = \delta_{C_1}(p + b_0, q)$. Applying the translation property of proximal operators and the Moreau identity, we have:

$$\text{prox}_{f_1^*}^{\sigma_1}((\bar{p}, \bar{q})) = (\bar{p}, \bar{q}) - \sigma \left( \text{prox}_{\delta_{C_1}}^{\sigma_1^{-1}} \left( \left( \frac{\bar{p}}{\sigma_1} + b_0, \frac{\bar{q}}{\sigma_1} \right) \right) - b_0 \right).$$

Here the proximal operator of $\delta_{C_1}$ is the epigraphical projection of leaky ReLU, which is given by:

$$\text{proj}_{C_1}(\bar{p}, \bar{q}) = \begin{cases} (\bar{p}, \bar{q}) & \text{if } h_1(\bar{p}) \leq \bar{q} \\ (\frac{\bar{p}+\bar{q}}{2}, \frac{\bar{p}+\bar{q}}{2}) & \text{if } |\bar{q}| \leq \bar{p} \\ (\frac{\bar{p}+\alpha\bar{q}}{1+\alpha^2}, \frac{\alpha(\bar{p}+\alpha\bar{q})}{1+\alpha^2}) & \text{if } \bar{q} \leq \alpha\bar{p} \text{ and } \bar{p} \leq -\alpha\bar{q} \\ (0, 0) & \text{otherwise} \end{cases}.$$

where $h_1$ denotes the leaky ReLU function with negative slope $\alpha$.

Consider $f_2(\mathbf{w}) = \gamma \mathbf{W}_2 h_2(\mathbf{w} + \mathbf{b}_1)$. Since $f_2$ is separable, we can first consider the simpler 1D function $\tilde{f}_2(w) = a \max(w + b, 0)$, its proximal operator can then be easily computed. Applying Moreau identity once again, we have:

$$\left[ \text{prox}_{f_2^*}^{\sigma_2}(\bar{w}) \right]_i = \begin{cases} [\gamma W_2]_i & \text{if } [\bar{w} + \sigma b_1]_i > [\gamma W_2]_i \\ 0 & \text{if } [\bar{w} + \sigma b_1]_i < [\gamma W_2]_i \\ \bar{w}_i & \text{if } 0 \leq [\bar{w} + \sigma b_1]_i \leq [\gamma W_2]_i \end{cases}.$$

The proximal operator of the $L^1$ data fidelity is given by the pointwise soft shrinkage function:

$$\left[\mathrm{prox}_{\lambda\|\cdot - y\|_2}^{\tau}(\bar{x})\right]_i = \begin{cases} \bar{x}_i - \tau\lambda & \text{if } \bar{x}_i - y_i > \tau\lambda \\ \bar{x}_i + \tau\lambda & \text{if } \bar{x}_i - y_i < -\tau\lambda \\ y_i & \text{otherwise} \end{cases}.$$

For the Kullback–Leibler divergence, we let $f_0(\mathbf{w}) = \mathbf{1}^T(\mathbf{w} - \mathbf{y} + \mathbf{r}) + \mathbf{y}^T \log\left(\frac{\mathbf{y}}{\mathbf{w}+\mathbf{r}}\right)$. The proximal operator of its conjugate can be given by (Chambolle et al., 2018):

$$\left[\mathrm{prox}_{f_0^*}^{\sigma_0}(\bar{\mathbf{w}})\right]_i = \frac{1}{2}\left(\bar{\mathbf{w}}_i + 1 + \sigma_0\mathbf{r}_i - \sqrt{(\bar{\mathbf{w}}_i - 1 + \sigma_0\mathbf{r}_i)^2 + 4\sigma_0\mathbf{y}_i}\right)$$

### C.3 STEP-SIZE SELECTION SCHEME

For the denoising and the inpainting experiments, we incorporate the data fidelity as $g$, and consider the operator:

$$\mathbf{K} = \begin{pmatrix} \mathbf{V}_0 & \mathbf{0} \\ \mathbf{0} & \mathbf{I} \\ \mathbf{0} & \mathbf{W}_1\mathbf{P} \end{pmatrix}.$$

Then $\mathbf{S}^{1/2}\mathbf{K}\mathbf{T}^{1/2}$ is given by

$$\mathbf{K} = \begin{pmatrix} \sqrt{\sigma_1\tau_1}\mathbf{V}_0 & \mathbf{0} \\ 0 & \sqrt{\sigma_1\tau_2}\mathbf{I} \\ 0 & \sqrt{\sigma_2\tau_2}\mathbf{W}_1\mathbf{P} \end{pmatrix}.$$

To study the convergence condition we compute

$$\begin{aligned}
\|\mathbf{S}^{1/2}\mathbf{K}\mathbf{T}^{1/2}\mathbf{u}\|^2 &= \sigma_1\tau_1\|\mathbf{V}_0\mathbf{x}\|^2 + \sigma_1\tau_2\|\mathbf{z}\|^2 + \sigma_2\tau_2\|\mathbf{W}_1\mathbf{P}\mathbf{z}\|^2 \\
&\leq \sigma_1\tau_1\|\mathbf{V}_0\|^2\|\mathbf{x}\|^2 + \sigma_1\tau_2\|\mathbf{z}\|^2 + \sigma_2\tau_2\|\mathbf{W}_1\mathbf{P}\|^2\|\mathbf{z}\|^2 \\
&= \sigma_1\tau_1\|\mathbf{V}_0\|^2\|\mathbf{x}\|^2 + (\sigma_1\tau_2 + \sigma_2\tau_2\|\mathbf{W}_1\mathbf{P}\|^2)\|\mathbf{z}\|^2.
\end{aligned}$$

By choosing the step-sizes as in (10), we have $\sigma_1\tau_1\|\mathbf{V}_0\|^2$, $\sigma_1\tau_2 + \sigma_2\tau_2\|\mathbf{W}_1\mathbf{P}\|^2 \leq 1$. In all the experiments, we computed the respectively norms using power methods.

For the CT experiment, we incorporate the data fidelity as $f_0$, and consider the operator:

$$\mathbf{K} = \begin{pmatrix} \mathbf{A} & \mathbf{0} \\ \mathbf{V}_0 & \mathbf{0} \\ \mathbf{0} & \mathbf{I} \\ \mathbf{0} & \mathbf{W}_1\mathbf{P} \end{pmatrix}.$$

Then $\mathbf{S}^{1/2}\mathbf{K}\mathbf{T}^{1/2}$ is given by

$$\mathbf{K} = \begin{pmatrix} \sqrt{\sigma_0\tau_1}\mathbf{A} & \mathbf{0} \\ \sqrt{\sigma_1\tau_1}\mathbf{V}_0 & \mathbf{0} \\ 0 & \sqrt{\sigma_1\tau_2}\mathbf{I} \\ 0 & \sqrt{\sigma_2\tau_2}\mathbf{W}_1\mathbf{P} \end{pmatrix}.$$

Similarly, we compute

$$\begin{aligned}
\|\mathbf{S}^{1/2}\mathbf{K}\mathbf{T}^{1/2}\mathbf{u}\|^2 &= \sigma_0\tau_1\|\mathbf{A}\mathbf{x}\|^2 + \sigma_1\tau_1\|\mathbf{V}_0\mathbf{x}\|^2 + \sigma_1\tau_2\|\mathbf{z}\|^2 + \sigma_2\tau_2\|\mathbf{W}_1\mathbf{P}\mathbf{z}\|^2 \\
&\leq \sigma_0\tau_1\|\mathbf{A}\|^2\|\mathbf{x}\|^2 + \sigma_1\tau_1\|\mathbf{V}_0\|^2\|\mathbf{x}\|^2 + \sigma_1\tau_2\|\mathbf{z}\|^2 + \sigma_2\tau_2\|\mathbf{W}_1\mathbf{P}\|^2\|\mathbf{z}\|^2 \\
&= (\sigma_0\tau_1\|\mathbf{A}\|^2 + \sigma_1\tau_1\|\mathbf{V}_0\|^2)\|\mathbf{x}\|^2 + (\sigma_1\tau_2 + \sigma_2\tau_2\|\mathbf{W}_1\mathbf{P}\|^2)\|\mathbf{z}\|^2.
\end{aligned}$$

By choosing the step-sizes as in (15), we have $\sigma_0\tau_1\|\mathbf{A}\|^2 + \sigma_1\tau_1\|\mathbf{V}_0\|^2$, $\sigma_1\tau_2 + \sigma_2\tau_2\|\mathbf{W}_1\mathbf{P}\|^2 \leq 1$.

# D    COMPARISON WITH SMOOTHED PROBLEM: IMAGE INPAINTING

In this section, we consider a smoothed version of our problem and compare the subgradient methods applied to this smoothed problem with the proposed method on the same image inpainting task as in Section 4.2. We consider the following smoothed approximation to ReLU:

$$
\tilde{h}_1(x) = \begin{cases} 0 & \text{if } x \leq 0, \\ \frac{x^2}{2\nu}, & \text{if } 0 < x < \nu, \\ x - \frac{\nu}{2}, & \text{otherwise}, \end{cases}
$$

where $\nu$ denotes a smoothing parameter. Similarly, we consider the smoothed approximation to leaky ReLU given by $\tilde{h}_2(x) = \kappa x + (1 - \kappa)\tilde{h}_1(x)$, where $\kappa$ corresponds to the negative slope of leaky ReLU.

The subgradient methods are applied to solve the following problem:

$$
\min_{\mathbf{x},\mathbf{z}} \frac{1}{2}\|\mathbf{A}\mathbf{x} - \mathbf{y}\|_2^2 + \gamma \tilde{R}_\theta(x), \tag{16}
$$

where $\tilde{R}_\theta(\mathbf{x}) = \mathbf{W}_2\tilde{h}_2(\mathbf{W}_1\mathbf{P}\mathbf{z} + \mathbf{b}_1)$ with $\mathbf{z} = \tilde{h}_1(\mathbf{V}_0 x + \mathbf{b}_0)$. The weights $\boldsymbol{\theta}$ are kept the same as those of the pre-trained model as in Section 4.2.

**Parameters:** We set the smoothing parameter $\nu$ as 0.01 and set $\gamma = 0.1$. For SM-C, we select the step-sizes from $\{0.5, 1, 1.5, 2\}$. For SM-D, the initial step-sizes are chosen from $\{10, 30, 50, 60\}$.

**Results:** Figure 10 compares the energy and PSNR plots with the smoothed version of the problem. While this smoothed formulation approximates the original problem, the subgradient methods behave similarly to their performance in the original setup, showing comparable trends in step-size choices, objective values, and PSNR values. For instance, the diminishing step-size strategy still fails to improve convergence speed, and both subgradient methods are still significantly slower compared to the proposed method in reducing the objective value, showing that smoothing does not improve the convergence speed for subgradient methods. Figure 11 presents comparisons of reconstructions with that of the smoothed problem, which are visually similar to those obtained from the original problem.

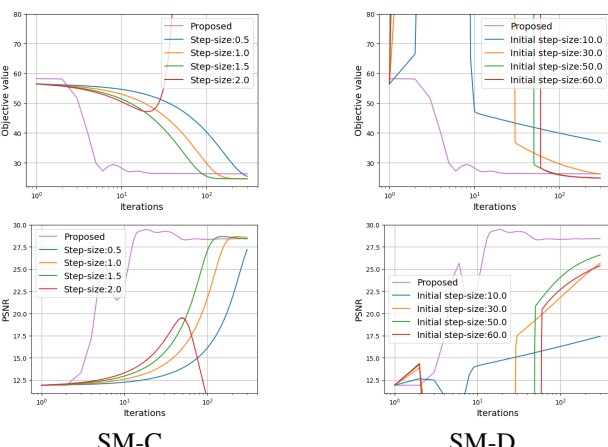

SM-C                           SM-D

Figure 10: Inpainting: Comparison to subgradient methods.

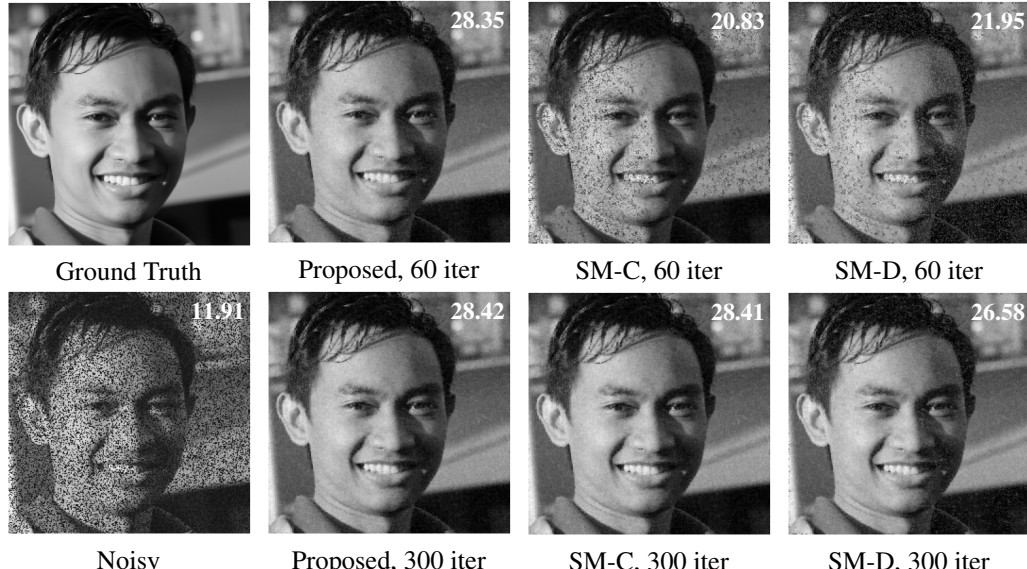

Figure 11: Inpainting: Visual comparison of reconstructions, with PSNR shown at top right corner.

