# OpenReview forum: "A primal-dual algorithm for variational image reconstruction with learned convex regularizers"
_ICLR.cc/2025/Conference — Submitted to ICLR 2025_

### Official Review · Reviewer_2wPy · 2024-10-31

**Soundness:** 4
**Presentation:** 3
**Contribution:** 2
**Rating:** 5
**Confidence:** 4

**Summary:**

This paper considers a classical fidelity + regularization model for image reconstruction, where the regularizer is an Input Convex Neural Network (ICNN). This construction, together with the choice of a fidelity term as a convex function, make the objective function of the optimization problem convex. However, the nested structure of such regularizer make the problem hard to optimize, therefore the authors consider an equivalent formulation where the nested structure is removed by inserting a constraint to the optimization problem. However, due to the non-convexity of the feasible set, the problem becomes non-convex. To avoid this issue, the authors propose to consider instead the epigraph of the feasible set, obtaining an optimization problem which is equivalent to the original optimization problem, while being convex. This allows the authors to use the fast and stable Primal-Dual (PD) algorithm from [Chambolle, Pock; 2011].

**Strengths:**

The paper is well-written and the math is mainly correct. The results shows that the proposed method is superior than the algorithms to which the authors compared, showing a small, but non-zero, improvement in the state-of-the-art to which this paper refers.

**Weaknesses:**

I believe the idea is not original enough to be published on a big conference such as ICLR. Indeed, to my understanding, the proposed model is obtained by combining the known and widely used ICNN-based regularizer (see, e.g. Learning Convex regularizers for inverse problems; Mukherjee et al.; 2021), with a known technique to remove the nested structure (Carreira, Perpinan; 2014) and the known epigraphical projection (Chierchia et al.; 2015). The problem is then solved through the application of the classical Primal-Dual optimization algorithm, widely used to solve convex non-smooth optimization problems. All of these methods appear, to the reviewer, to be combined with no major improvements or adaptations with respect to the paper from which they are taken. While this could not be a major issue for most journals and conferences, I believe this is not enough novelty to be accepted and published on a conference such as ICLR. Clearly, I am open to change my mind if I understood it wrong.

Anyway, I also found some aspects of the paper which I suggest to improve:

-	The most relevant criticism is on the experimental section. In particular, the authors compare their ICNN-based regularization model solved by PD algorithm against the same model solved by other Subgradient Methods (SM). However, I believe that the optimization algorithm employed is the least innovative aspect of the paper, in particular when compared to the choice of the learned regularizer. I suggest the authors to compare their method with other regularization methods such as Total Variation, Wavelets, or any other learned regularization method. Moreover, I kindly ask the authors to explain their rationale or focusing on optimization algorithm comparisons rather than regularizer comparisons.
-	The method on which the authors compare (i.e. the SM-C and SM-D methods) are not introduced nor cited. In particular, I did not find any reference to which algorithms have been employed specifically, and which choice of the parameters has been employed. Could you please provide them?
-	Be more specific in the proof of Theorem 3, you say that the two formulations are equivalent, but you only show that their minimum is the same. For two optimization problems to be equivalent, they are supposed to have the same minimum points, i.e. the two minima are achieved at the same value of $x$. This does not necessarily hold if they have the same minima. For example, the functions $f(x) = x^2$ and $g(x) = (x-1)^2$, both have the same minimum (i.e. 0), but their minimum points is $x=0$ for $f(x)$, $x=1$ for $g(x)$. Therefore, they are not “equivalent”.

**Minor Comments.**

-	In line 66, “step-size selection” should be erased.
-	In line 69-70, there is a repetition of “remove”.
-	In Assumption 2, $D$ does not directly depend on $x$, while it depends on $x$ through $A$. I suggest to explicitly say that “$D(Ax, y)$ is convex in $x$”.
-	In Theorem 3, specify is the “min” in $w_{L-1}$ is a minimum or an inf, since the notation is changed through the paper.
-	In line 241, add a citation to SM-D and SM-C methods.
-	In equation (10), how do you compute the matrix norms in the algorithm?
-	In Figure 3, when the legend says “Proposed”, to which parameter choice it refers? I also suggest to add a grid on the plot background of Figure 3 to improve readability.
-	In line 314, “3% Gaussian noise”, what does it mean? You add Gaussian noise with standard deviation of 0.03? The norm of the noise is 3% of the norm of the data?
-	In the Computed Tomography (CT) experiment, the amount of information measured is too high, making the problem “easy” to solve. I suggest testing more extreme geometries such as 60-90 angles, instead of 200, to show how the proposed method behaves in other scenarios.

**Questions:**

I included a few questions in the "Weakness" section.

---

> ### Author Response · Authors · 2024-11-25
> **Response to reviewer 2wPy**
>
> #### **Weaknesses**
>
> **1. Originality of the Idea**
> We understand your concerns about the perceived originality of our work. While techniques for removing nested structures have been explored, such as by Carreira-Perpinan & Wang (2014), these methods were designed for training neural networks and relied on penalty terms in reformulations, which are not equivalent to the original problem.
>
> Our approach is novel in several key aspects:
> - We integrate these methods into a unified framework tailored for solving the *inference problem* (P) with ICNNs.
> - We preserve convexity, leveraging the structure of ICNNs.
> - Crucially, we **prove the equivalence** of our reformulation (P1) and the original problem (P), a significant theoretical contribution that prior works do not offer.
> - Finally, we address the challenge of nonsmoothness by enabling the use of primal-dual methods, overcoming the limitations of subgradient and proximal-gradient methods.
>
> **2. Focus on Optimization Algorithm Comparisons**
> Our focus on optimization algorithms rather than regularizers stems from the following rationale:
> - The primary contribution of this work lies in developing an efficient algorithmic framework to address nonsmooth optimization problems involving ICNNs.
> - Existing methods struggle with challenges such as the slow convergence of subgradient methods and the infeasibility of computing exact proximal operators for ICNN-based regularizers. Our method directly addresses these issues.
>
> While comparisons with other regularizers, such as Total Variation or Wavelets, are indeed relevant, they diverge from the main objective of this work. Exploring regularizer comparisons is an exciting avenue for future research, but we believe the optimization framework is indeed the most innovative aspect of this paper.
>
> **3. Introduction of SM-C and SM-D Methods**
> Thank you for pointing out the lack of details regarding the subgradient methods (SM-C and SM-D). In the revised version, we have included a detailed description of SM-C and SM-D, including their formulations and parameter choices. References are also included.
>
> **4. Proof of Theorem 3**
> We appreciate your observation regarding Theorem 3. In the revised manuscript, we have expanded the proof to demonstrate the original problem (P) and the reformulated problem (P1) are equivalent. This provides a more rigorous and complete justification of their equivalence.
>
> ---
>
> #### **Minor Comments**
>
> **1. Matrix Norm Computation**
> We compute the matrix norms using the power method. This has been clarified in the revised text.
>
> **2. Gaussian Noise Specification**
> The "3% Gaussian noise" refers to adding noise with a standard deviation of 0.03. We have included this clarification in the revised manuscript.
>
> **3. "Easy" CT Experiment**
> Your comment about the CT experiment is well-taken. The chosen geometry already leads to a forward operator with a non-trivial null space. Since our primary focus is on the optimization algorithms rather than the effectiveness of the regularizer, we believe that 200 angles provide a sufficient basis for our experiments.
> From an optimization perspective, more extreme geometries could introduce additional challenges, such as ill-conditioning, which may highlight further advantages of our approach. This is an interesting direction for future work.

---

> > ### Comment · Reviewer_2wPy · 2024-11-26
> >
> > Thanks to the authors for their reply. The updated proof looks correct to me. I will raise my evaluation from 3 to 5. However, I believe this paper addresses an audience that does not align completely with ICLR. For this reason, even if I believe the work is interesting and worth publishing, I will stick with 5 points.

---

### Official Review · Reviewer_JEEV · 2024-11-01

**Soundness:** 4
**Presentation:** 3
**Contribution:** 2
**Rating:** 5
**Confidence:** 5

**Summary:**

In this work the authors consider a primal-dual algorithm for solving composite optimisation problems where the regularisation term is parameterised by a non-smooth ICNN. In fact, the authors first present a more general input-convex type reformulation of the regularisation which is more general than the standard ones and then, using epigraphical projection reformulation, rewrite in a convex way the training problem, solving it with a tailored primal-dual algorithm. Numerical tests are showed.

**Strengths:**

The paper is very nicely written, concise and well-structured.
It provides a good overview of the state-of-the-art approaches existing in the fields of image reconstruction with network-parametrised regularisation models, a good presentation of the fremework of ICNN and the nice idea of convexifying the (in principle) more general reformulation introduced by the authors via epigraphical projections, which is, essentially, a lifting procedure making the relaxed problem convex and amenable to be solved by, e.g., primal-dual algorithms.
Quite a few numerical results are presented.

**Weaknesses:**

My main concern is the only comparison with subgradient methods in terms of computational efficiency and the resulting claim that the proposed algorithm improves upon it. This is clearly the case, but the reason is that subgradient method is well-known to be very slow ( O(1/\sqrt{k}) w.r.t. to the function values! I think that something that the authors should do is including a comparison with smooth solvers applied to slightly smoothed versions of their architectures. That would allow you to use gradient-type algorithms and make comparisons w.r.t. to O(1/k) convergent approaches, which I believe is more representative of the speed you obtain.
The other weakness of this work si that the authors introduce a more general theoretical framework in Section 2  which, in practice, is not really used later on in the experiments. The framework proposed, indeed, is nice but hardly related to the standard linear + non-linear nature of NNs, so the authors indeed simply apply it to ICNN, as expected. I am not sure then that this relies provides anything interesting from a theoretical view point.
The other major limitation is that while nice from a theoretical viewpoint, ICNNs perform worse than less-structured networks. It would be nice to show a baseline comparison w.r.t. to non-convex or, better, weakly-convex regularisers on which, I believe, analogous considerations could be made but with much better performance.
Also, while the authors declare explicitly their interest into imaging applications starting from the title, I think that some further tests to more ML-related applications showing the interest of ICNN there (if any) would be appreciated.

**Questions:**

- I think the author should at least talk about proximal-gradient methods to address non-smooth problems. Primal-dual algorithms are not the only algorithms used for this.
- The sentence "the above problem is non-convex despite the objective is convex" is very foggy. I would be more precise here and relate the underlying non convexity to the constraint considered.
- The authors should add a comparison with gradient-type algorithms used to solve smoothed version of their problem for making conclusions about the computational gain in comparison also to a smoothed version of the regulariser considered. Somethimes a slight smoothing is better when balancing the computatonional/reconstruction improvements.
- I am not sure I understand the idea of showing results in early iterations where the training process is far from having converged.
- The subgradient algorithm should be written. Moreover, for each problem considered, the subgradients should be written and reported, at least, in the Supplementary material.
- A further comparison with less-structured regularisation network (non-convex or weakly-convex) should be added to position your results w.r.t. to the state-of-the-art.

---

> ### Author Response · Authors · 2024-11-25
> **Response to reviewer JEEV**
>
> #### **Weaknesses**
>
> **1. Comparison with Subgradient Methods and Computational Efficiency**
> We appreciate your concern about the reliance on subgradient methods for comparison. As noted, the optimization problem (P) is nonsmooth, and subgradient methods are widely used in the literature for such problems (e.g., Mukherjee et al., 2021). One of the motivations of our work is the slow convergence of subgradient methods. While smooth activations can alleviate non-smoothness to some extent, many commonly used activations remain nonsmooth. Our proposed method is designed to handle such cases effectively, offering flexibility and improved computational efficiency.
>
> Additionally, the non-smoothness in our problem does not solely stem from the regularizer but also from the data fidelity, as shown in the salt-and-pepper denoising example. Proximal gradient methods, while effective for many nonsmooth problems, are not practical in this case because the proximal operator of the regularizer cannot be computed exactly. These challenges are precisely why we developed the proposed method, which reformulates and addresses the optimization problem in a computationally efficient manner.
>
> **2. General Theoretical Framework in Section 2**
> We agree that our theoretical framework is presented in a more general setting than what is applied in our experiments. The intention here is to establish a broader mathematical foundation that can support extensions of our method in future research. For instance, the results presented can accommodate more complex architectures like ResNets, and we aim to explore this potential in subsequent work.
>
> **3. Comparison with Non-Convex or Weakly Convex Regularizers**
> Thank you for suggesting a comparison with less-structured regularizers. Our work specifically focuses on the convex regime due to its theoretical advantages, such as stability, convergence guarantees, and global minimizers. Nevertheless, we recognize the value of exploring non-convex and weakly convex regularizers and plan to consider this in future studies.
>
> **4. Tests on ML-Related Applications**
> We appreciate your suggestion to evaluate our method in machine learning-related applications. While this paper focuses on image restoration tasks, we agree that investigating the utility of ICNNs in broader machine learning scenarios could provide valuable insights. We hope to pursue this direction in future work.
>
> ---
>
> #### **Questions**
>
> **1. Proximal-Gradient Methods for Non-Smooth Problems**
> Proximal-gradient methods are indeed valuable tools for nonsmooth problems. However, as highlighted in the paper, the proximal operator of the regularizer in our context cannot be computed exactly. This limitation renders proximal-gradient methods impractical for our problem. To clarify this point further, we have revised the introduction and moved the relevant discussion to the "Challenges and Motivations" section.
>
> **2. Ambiguity in "The above problem is non-convex despite the objective is convex"**
> Thank you for identifying this unclear statement. We have revised it to explicitly state that the feasible set of the optimization problem is non-convex, even though the objective function itself is convex.
>
> **3. Comparison with Gradient-Type Algorithms on Smoothed Versions**
> We acknowledge the importance of comparing with gradient-type algorithms applied to smoothed versions of the problem. While non-smoothness may also arise from the data fidelity term, we have included additional comparisons with subgradient methods applied to smoothed versions of the regularizer in the inpainting task.
>
> **4. Showing Results During Early Iterations**
> We understand the confusion regarding results shown at early iterations. To clarify, the results presented are not during the training process. Instead, the regularizer is pre-trained, and we show the reconstructions (variable \( x \)) at different iterations of the optimization algorithms. This approach visually demonstrates the faster convergence and superior reconstruction quality achieved by our method compared to subgradient methods. Subgradient methods require significantly more iterations to achieve similar results, underscoring the efficiency of our approach.
>
> **5. Writing the Subgradient Algorithm**
> We agree that providing detailed descriptions of the subgradient algorithm would be helpful for reproducibility. We have included these details in the supplementary material.
>
> **6. Comparison with Less-Structured Regularization Networks**
> As discussed earlier, our focus is on the convex regime to leverage its theoretical guarantees. While comparing with non-convex or weakly convex regularizers is intriguing, it is outside the scope of this paper. We believe this is a valuable direction for future research.

---

> > ### Comment · Reviewer_JEEV · 2024-11-26
> > **Thanks for the revision but I am still totally sure about the impact of this work**
> >
> > I appreciate the time taken by the authors to carefully substantiate their work w.r.t. the many comments received by the reviewers. I understand that the field of application of the approach proposed is here voluntarly restrained to imaging problems and convex problems but paves the way to further interesting extensions which could have or not been addressed in this work.
> > I am still a bit skeptical about the overall impact that this work may have in comparisons to other existing approaches that to me were a bit overlooked here. I appreciate that the authors added a final experiment on the use of subgradient method on a smooth problem (it is therefore a gradient method) showing improvements but this should have possibly be added to all the experiments in the main body of the paper.  I understand that nonsmoothness may come not only from non-smooth networks parametrizing the regularizer, but also on the data terms (as in the case of l1), which would have deserved a comparison after smoothing too.
> > I will maintain my score as I think that this work is good but better suited for the more specific imaging community rather than the ICLR one.

---

### Official Review · Reviewer_qRw1 · 2024-11-02

**Soundness:** 3
**Presentation:** 3
**Contribution:** 2
**Rating:** 5
**Confidence:** 3

**Summary:**

In this paper, the authors propose a novel, more general, formulation of input-convex neural networks (ICNN). The main contribution to the paper consists in relaxing the usual constraint that layerwise functions should be increasing with a mere convexity constraint. This more general framework encompasses the traditional case, but allows for more general neural network architectures. The authors showcase their new architecture on various imaging problems: salt-and-pepper denoising, image inpainting and CT reconstruction.

**Strengths:**

Overall, this is a well written paper that is both original and interesting.

1. The paper is original. It goes against the current trend of always larger architectures, and makes interesting links with convex optimization.
2. The paper is mathematically sound.
3. The paper is well written and very clear.

**Weaknesses:**

The main weakness of the paper is the experimental evaluation. As such, it is difficult to put the work in context of other works due to the lack of relevant baselines (see details in the "questions" section).

1. There are no baselines to which the authors compare their approach in the experimental section.
2. The constraints on the architecture seem quite strong (weight positivity, see comments below).
3. Differences with respect to existing works (PnP, ICNNs) could be further clarified (see comments below).

**Questions:**

**Main comments**
1. The proposed framework does not limit the number of layers $L$ in the ICNN. However, in the experimental part, the number of layers is set to a maximum of 2 (see equations (8) and (11)). Would a larger number of layers yield better results? Is there a reason for such a low number of layers in practice?
2. I understand that it is not possible to implement all methods and that this paper is quite theory oriented. However, I think that the paper would greatly benefit from (by order of importance) (1) comparisons with other baselines (in particular, the Goujon et al reference proposed by the authors, or even comparisons with much simpler convex models, e.g. wavelets/TV) (2) experiments in color images (3) experiments on Gaussian denoising.
3. How does the proposed method compare to plug-and-play algorithms? For instance, it would seem a natural choice to train $R_\theta$ as a Gaussian denoiser. However, the authors train the model in an unrolled fashion, meaning that a new regularizer needs to be learned for new problems. Could the authors comment on that?
4. The method has 2 major differences with the common architectures for image restoration. (a) first, the fact that the model is tied to an image size; how can the architecture be used for images of sizes different than 256$\times$256? (b) second, the positivity of the convolutional weights seems to be a very strong constraint. Did the authors try to relax this assumption, even-though it departs from the convex setting?
5. Is it correct to state that (8) and (11) do not fit in the standard ICNN framework because $\delta_{C_1}$ is not increasing? I would encourage the authors to strongly state how their new framework differs from that of Amos et al. at the level of the experiments.
6. Some exciting application of this work would be in image quality assessment (IQA). In essence, one can see $R_\theta$ as a measure of the quality of an image. Did the authors consider such application for their model?

**Minor comments**
1. Figure 1 is not very clear: $0.5w_1 + 0.5w_2$ could be put below the blue "x" where it is located, similarly as is done for w1 and w2.
2. Line 125, $z_i$ should be replaced with $\omega_i$ in the expression of $\phi_i$.

---

> ### Author Response · Authors · 2024-11-25
> **Response to reviewer qRw1**
>
> ### Weaknesses
>
> #### **No Baselines for Comparison**
> We appreciate your observation regarding the lack of baselines in the experimental section. Given the nonsmooth nature of the optimization problem, the literature typically employs subgradient methods, which we believe serve as an appropriate baseline. While proximal gradient methods are effective for many non-smooth problems, as mentioned in the paper, the proximal operator of the regularizer in our framework cannot be computed exactly. Consequently, proximal methods have not been commonly employed for such problems.
>
> #### **Strong Constraints on the Architecture**
> The constraints on the architecture, such as convexity and positivity, arise from the need to ensure the ICNN's convexity (see Amos et al., 2017). These constraints are intrinsic to the ICNN framework. While exploring alternative architectures could be valuable, this lies beyond the scope of our current work, which focuses on the optimization perspective.
>
> #### **Clarifying Differences with Existing Works (PnP, ICNNs)**
> Our framework addresses the specific optimization problem (P) using a pre-trained ICNN as a regularizer. This fundamentally differs from Plug-and-Play (PnP) approaches, which replace the proximal operator with a denoiser, or unrolling methods, which train the regularizer in an unrolled fashion. We provide further clarifications regarding these distinctions in response to specific questions below.
>
> ---
>
> ### Questions
>
> #### **1. Would a larger number of layers yield better results?**
> We agree that increasing the number of layers may enhance reconstruction quality. However, our focus is not on improving the reconstruction quality by varying architectures but on solving the optimization problem (P) efficiently with a pre-trained regularizer. Please also refer to the general response for additional context regarding our choice of a low number of layers for simplicity and proof of concept.
>
> #### **2. Suggestions for Comparisons and Additional Experiments**
> We appreciate your suggestions regarding comparisons with baselines, experiments with color images, and Gaussian denoising. While these are important topics in the broader context of image reconstruction, they are outside the scope of this paper. Our focus is on the optimization framework and its theoretical guarantees. Comparisons or experiments that target specific reconstruction tasks are potential extensions for future work.
>
> #### **3. Comparison with Plug-and-Play Algorithms**
> Plug-and-Play (PnP) algorithms differ fundamentally from our approach. PnP typically uses proximal gradient methods where the proximal operator is replaced with a denoiser. In contrast, our approach employs a pre-trained ICNN as a regularizer. Additionally, our framework does not involve unrolled training. The regularizer is trained independently using an adversarial framework, and it is then used to solve the variational problem in our framework. While comparisons with PnP and unrolling methods are intriguing, they are not the primary focus of this work.
>
> #### **4. Model Constraints**
> The choice of fully connected layers is purely for illustrative purposes, demonstrating our ability to work with various common components of a network. To enable the method to handle images of varying sizes, ICNNs which replace fully connected layers with convolutional layers can be used.
>
> The constraint on weight positivity ensures the convexity of the regularizer, which is crucial for computing global minimizers and leveraging convex optimization theory for guarantees of stability and convergence. While non-convex regularizers, such as weakly convex ones, could be considered in future work, this paper focuses on the benefits of convexity.
>
> #### **5. Suitability of Equations (8) and (11)**
> Equations (8) and (11) represent the unconstrained versions derived from our reformulation. We would like to clarify that our architecture adheres to the standard ICNN framework proposed by Amos et al. (2017). The indicator function in our formulation serves to encode inequality constraints and is not an activation function, so it does not need to be non-decreasing. Our contribution lies in reformulating problem (P) into (P1), with proven equivalence, rather than introducing a new ICNN architecture.
>
> #### **6. Applications in Image Quality Assessment (IQA)**
> Thank you for highlighting the potential application of our work in image quality assessment (IQA). While we have not explored this in the current paper, it is indeed a promising direction for future research.
>
> ---
>
> ### Minor Comments
>
> #### Line 125, $z_i$ should be replaced with $\omega_i$  in the expression of $\phi_i$.
> We respectfully believe that $z_i $ should remain in line 125. In the ICNN architecture, $\omega_i$ represents the outputs of all previous layers, while $z_i$ specifically denotes the output of the last layer. The formulation relies only on $z_i$, as outlined in (EQ).

---

> > ### Comment · Reviewer_qRw1 · 2024-12-02
> >
> > I thank the authors for their detailed response. Unfortunately, I am not sure that the proposed modifications by the authors adress my concerns, that is the experimental validation of the proposed approach. Let me detail my points below.
> >
> > 1. In the rebuttal, the authors claim that "The key focus of our work lies in the efficiency of optimization algorithms, rather than improving reconstruction quality through varying the choice of regularizer." A main issue with the paper in its current form is that the experiments are oriented towards image restoration problems, hence my desire for comparisons with other restoration methods. To be clear, I suggest this as an "easy" fix for the paper in its current form, solely requiring more baselines on image restoration problems.
> >
> > If instead the authors aim to focus on a simple problem as they claim in the rebuttal, with shallow models - this is totally fine - then I would expect other, simple experiments that would demonstrate the validity of the approach on other problems and imaging problems should only appear as an instance of problems among others, or populating Table 1 with more method - in fact, Table 1 is the most important results part if the authors focus mainly on the method. For instance, the authors could toy 2d constrained problems, with more detailed convergence bounds, showing that experiments match theory, etc... (why not consider toy classification problems either?) This naturally requires more work, but I think this is necessary to convey the desired message of the authors, because at the moment, the authors propose a theoretical paper in the first half of the paper, and an imaging paper in the second half of the paper. (the authors may disagree with this, but this is how it reads for me).
> >
> > 2. I appreciate the authors' clarifications in the manuscript. A point that I struggle to understand is the necessity for the constraints in (P1). In practice, are these constraints not automatically satisfied due to ICNN architectural design (e.g. Assumption 1)? If yes, are methods like Adam able to solve problem (P1)? In the case where this is possible, adding this as a comparison for the baselines would be helpful.

---

### Official Review · Reviewer_mQ6N · 2024-11-03

**Soundness:** 3
**Presentation:** 3
**Contribution:** 2
**Rating:** 5
**Confidence:** 5

**Summary:**

The paper addresses the optimization problem in a data-driven variational reconstruction framework where a neural network-based regularizer, specifically an input-convex neural network (ICNN), is used. The authors propose a reformulation of the problem that removes the nested nature of a network structure. By linking this reformulation to epigraphical projections of activation functions, they convert the problem into a convex optimization format that is suitable for efficient solution via a primal-dual algorithm.

**Strengths:**

- The authors propose a reformulation of the problem that removes the nested nature of a network structure. By linking this reformulation to epigraphical projections of activation functions, they convert the problem into a convex optimization format that is suitable for efficient solution via a primal-dual algorithm.

- The authors claim to remove the assumption that the activation function does not need to be non-decreasing.

- Numerical validation is presented using the learned convex regularizers.

**Weaknesses:**

- The authors claim that their framework does not require the activation function to be non-decreasing; however, the experiments only involve activation functions that are both non-decreasing and convex. Validation with activation functions that are convex but decreasing would strengthen this claim. For instance, how might the framework perform with functions like the negative exponential or negative logarithm functions? These are provided as illustrative examples.

- Additionally, a plot showing the landscape or level curves of the learned regularizer (neural network) would provide valuable visual insight. For instance, the authors can make use of 2D contour plots or 3D surface plots of the regularizer output for different input features. Also, analyzing how the landscape changes during training would help to visually validate the convexity of the neural network.

- Based on my previous comments some discussion about the limitations of the proposed framework is needed. For instance, would be interesting to discuss about the computational complexity for larger networks, potential issues with certain types of imaging problems.

**Some minor comments**

- I suggest the authors to change the notation across the manuscript. Vectors in boldface and matrices in capital boldface. Aligning with this standard notation will improve clarity and make the manuscript more consistent with conventions commonly used in the literature.

- I suggest the authors to define $\theta$ immediately after equation (EQ) to ease the reading.

- Since the authors prove the convexity of the built neural networks would be interesting and needed to plot the landscape/curve levels of the learned regularizer to have a visual understanding of it. Maybe using the reference below.

*Li, H., Xu, Z., Taylor, G., Studer, C., Goldstein, T. (2018). Visualizing the loss landscape of neural nets. Advances in neural information processing systems, 31.*

**Questions:**

- A primary limitation of this paper is that the learned convex regularizers are restricted to ReLU and leaky ReLU activation functions. Could the authors consider extending their framework to support a broader range of activation functions? For instance is it possible to extend the guarantees presented in this paper to the composition of convex functions? What would be the theoretical challenges in extending the framework to algebraic operations between convex functions?

- The neural networks examined in this study are relatively small in scale, raising an important question about scalability. How well does the proposed primal-dual algorithm perform when training larger networks?

- On top of my previous comment, the authors did not discuss if the studied networks size are enough for any regularized imaging problem? Could the authors discuss whether the network sizes explored are sufficient for a range of regularized imaging problems? Could the authors provide the computational complexity of the algorithm as the network size increases? Also, some experiments with networks of increasing depth and width could help to demonstrate scalability.

- In sparse image recovery, the regularizer must satisfy certain constraints beyond convexity to guarantee unique recovery (see Proposition 4.4 in the reference below). Could the authors provide further explanation on how the proposed learned regularizers address these requirements to ensure uniqueness?

*Woodworth, J., Chartrand, R. (2016). Compressed sensing recovery via nonconvex shrinkage penalties. Inverse Problems, 32(7), 075004*

---

> ### Author Response · Authors · 2024-11-25
> **Response to reviewer mQ6N**
>
> ### Weaknesses
>
> #### **Choice of Activation Functions**
> As noted, we followed Amos et al. (2017) by using non-decreasing and convex activation functions, such as ReLU and leaky ReLU. However, our theoretical framework is more general, as it relaxes the monotonicity requirement for the activation function $h_1$. Exploring a broader class of activation functions within our framework is an interesting direction for future research.
>
> #### **Plot of Landscape**
> We appreciate the suggestion to study the training landscape, which indeed could provide valuable insights. However, for this paper, our focus is on algorithmic efficiency and conceptual clarity. Visualizing the landscape might unintentionally shift the focus toward designing regularizers, which is not the primary goal of our work. We believe this direction is better suited for future studies.
>
> #### **Limitations, Complexity, and Potential Issues**
> We have addressed these concerns in detail in our responses to the specific questions below.
>
> ---
> ### Minor comments
> Thank you for the suggestions, we have changeg the notations in our revised manuscript, and we have defined after equation (EQ).
>
> ---
> ### Questions
>
> #### **1. A primary limitation of this paper is that the learned convex regularizers are restricted to ReLU and leaky ReLU activation functions. Could the authors consider extending their framework to support a broader range of activation functions? For instance is it possible to extend the guarantees presented in this paper to the composition of convex functions? What would be the theoretical challenges in extending the framework to algebraic operations between convex functions?**
>
> We believe that ReLU and leaky ReLU activation functions are representative, as they are commonly employed in practice. Our general formulation (EQ-G) offers a theoretical basis for considering more complex architectures, such as ResNets. Exploring different convex structures would be an interesting avenue for future research.
>
> #### **2. The neural networks examined in this study are relatively small in scale, raising an important question about scalability. How well does the proposed primal-dual algorithm perform when training larger networks?**
>
> The networks studied in this paper were intentionally kept small to serve as a proof of concept and simplify the training process. However, the proposed primal-dual algorithm is designed to handle larger network architectures efficiently. The simplicity of the updates—consisting primarily of subtraction and projection means that the method scales linearly with increasing network size. Exploring scalability with larger networks is an exciting direction for future work.
>
> #### **3. On top of my previous comment, the authors did not discuss if the studied networks size are enough for any regularized imaging problem? Could the authors discuss whether the network sizes explored are sufficient for a range of regularized imaging problems? Could the authors provide the computational complexity of the algorithm as the network size increases? Also, some experiments with networks of increasing depth and width could help to demonstrate scalability.**
>
> For this proof of concept, we believe the studied network sizes are sufficient, as evidenced by the visually satisfactory reconstruction results presented in the paper. That said, we agree that deeper networks could further improve reconstruction quality for more complex imaging problems.
>
> In terms of computational complexity, the proposed algorithm scales linearly with respect to the depth of the network. This is because the updates for the additional primal and dual variables involve only straightforward operations like subtraction or projection. We appreciate the suggestion to include experiments with networks of increasing depth and width; these would provide a compelling demonstration of scalability and are planned for future work.
>
> #### **4. In sparse image recovery, the regularizer must satisfy certain constraints beyond convexity to guarantee unique recovery (see Proposition 4.4 in the reference below). Could the authors provide further explanation on how the proposed learned regularizers address these requirements to ensure uniqueness?**
>
> If a unique solution is desired, we can incorporate a small $L^2$ penalty to the objective. This would make the problem strongly convex, thereby guaranteeing a unique solution to the optimization problem (P).

---

### Official Review · Reviewer_n7k3 · 2024-11-04

**Soundness:** 3
**Presentation:** 3
**Contribution:** 2
**Rating:** 5
**Confidence:** 4

**Summary:**

This paper introduces a novel relaxed formulation of Input-Convex Neural Networks (ICNNs), expressed as a convex minimization problem over intermediate activations. This approach is applied as a regularizer for image inverse problems. The authors solve the resulting convex optimization problem using the Primal-Dual Hybrid Gradient (PDHG) method. The ICNN regularizer is trained in a task-specific manner within an adversarial framework. The effectiveness of this approach is demonstrated on three image inverse problems: salt-and-pepper noise removal, image inpainting, and sparse-view computed tomography (CT) reconstruction.

**Strengths:**

- The paper is well-structured and articulates its contributions clearly, making it accessible and easy to follow for readers.
- The reformulation of the ICNN as a convex minimization problem, combined with its solution through the Primal-Dual Hybrid Gradient (PDHG) algorithm, is innovative and shows strong potential for practical applications.
- The formulation and theoretical backing of the ICNN as a convex minimization problem are solidly presented, providing a strong theoretical foundation for the proposed approach.
- While the focus is on image inverse problems, the proposed ICNN could potentially be extended to other fields, such as optimal transport, where a deep convex potential is beneficial, showing the paper's broad applicability.

**Weaknesses:**

- **Limited Comparison in Experiments**: The experimental section only compares the proposed method with a subgradient algorithm applied to the same objective function, which limits the scope of the evaluation. It would strengthen the findings if the method were compared against a standard ICNN approach or a similar architectures without constraints, as this would better illustrate the relevance and advantages of the proposed modifications.

- **Insufficient Training Details**: The paper lacks clarity on the training process for the ICNN-based regularization. Key aspects, such as the training methodology, hyperparameter choices, and datasets, are either under-explained or absent, which may impact the reproducibility and clarity of the results.

- **Rationale for Adversarial Training Choice**: The use of an adversarial framework for training the regularization introduces task dependency, which limits the general applicability of the approach. The rationale behind this choice is not sufficiently justified, especially given that adversarial training can be complex and potentially less efficient compared to alternative approaches. Exploring or explaining why more standard training schemes, such as training by denoising, were not pursued could improve the paper’s rigor and scope. Denoising-based training could make the regularizer more versatile and less sensitive to specific tasks, potentially offering broader applicability.

- **Clarity in Problem Motivation**: While the limitations of ICNNs are well discussed in the abstract, they are not sufficiently detailed in the introduction. Expanding on these limitations in the introduction would provide clearer context and stronger motivation for the proposed method.

- **Simplistic Architecture**: The proposed ICNN architecture is relatively simple, with only two layers. Recent works have explored deeper ICNN architectures that could improve performance in complex tasks, and the paper would benefit from a comparison or discussion on how a deeper architecture might impact results.

**Questions:**

- **Simplification of Training**: Classical ICNNs are known for being difficult and unstable to train due to stringent constraints. Does your proposed formulation help simplify the training process compared to traditional ICNNs? Is this formulation more flexible in practice, or does it still face similar training constraints?

- **Details of the Comparison**: In the experimental section, is the subgradient method used for comparison applied to the same objective function as the proposed ICNN formulation, or is it applied to a standard ICNN? Clarifying this would help in assessing the effectiveness and relevance of the comparison.

---

> ### Author Response · Authors · 2024-11-25
> **Response to reviewer n7k3:**
>
> ### Weaknesses
>
> #### **Limited Comparison in Experiments**
> As highlighted in the general response, our focus is on designing new algorithms for solving problem (P). Consequently, the only meaningful comparisons are between algorithms that address this specific problem. In our experiments, we compared our approach with what we consider the standard ICNN. If the reviewer believes another variant should be deemed standard, we would greatly appreciate clarification on this point. It is also worth noting that such comparisons are only meaningful within the convex setting.
>
> #### **Insufficient Training Details**
> We appreciate your feedback regarding the lack of clarity in the training process. In response, we have provided additional details about the adversarial training framework in the Appendix, ensuring greater transparency and reproducibility.
>
> #### **Rationale for Adversarial Training Choice**
> We acknowledge your concern about the choice of the adversarial training framework. As stated in the general response, our work primarily evaluates the performance of algorithms for solving problem (P) with the same regularizer, focusing on the convex regime associated with ICNN-based regularizers. The proposed method is independent of the training process, and adversarial training was selected because it is, in our view, the simplest approach for training an ICNN-based regularizer.
>
> #### **Clarity in Problem Motivation**
> Thank you for your comment. To address this concern, we have rewritten the introduction section and detailed the challenges associated with ICNN regularizers in Section 2. These revisions aim to clarify the motivation behind our proposed method and its importance.
>
> #### **Simplistic Architecture**
> We acknowledge your feedback regarding the simplicity of our architecture. As noted in the general response, this choice was intentional to focus on demonstrating the effectiveness of our framework for solving the optimization problem (P). While deeper architectures could potentially enhance performance, our current work prioritizes algorithmic efficiency and conceptual clarity. Exploring the impact of deeper architectures is a promising direction for future research.
>
> ---
>
> ### Questions
>
> #### **Does your proposed formulation simplify the training process compared to traditional ICNNs? Is this formulation more flexible in practice, or does it face similar training constraints?**
> Although our primary focus is solving the optimization problem (P), we believe that our framework could be extended to address training challenges in ICNNs. While our framework could be beneficial to the training process, a thorough exploration of this topic is beyond the scope of the current paper and remains an area for future investigation.
>
> #### **In the experimental section, is the subgradient method used for comparison applied to the same objective function as the proposed ICNN formulation, or is it applied to a standard ICNN?**
> Yes, all methods compared in the experiments solve the same optimization problem. Specifically, the subgradient method computes subgradients for the objective defined in problem (P), while our proposed method addresses this problem via a reformulation proven to be equivalent to (P). This equivalence, as demonstrated in Theorem 3, constitutes one of the key contributions of our work.

---

> > ### Comment · Reviewer_n7k3 · 2024-12-02
> > **Responds to authors**
> >
> > Thanks for the clarifications which improved the quality of the manuscript.  I think that the proposed method is original. I think that the paper is also in the scope of the conference. However, the limited experiments (very small scale architecture) and the limited comparisons with existing methods (beyond ICNNs) does not allow me to raise my score.

---

### Author Response · Authors · 2024-11-25
**General response to all reviewers:**

We would like to thank all the reviewers for your valuable feedback and insightful comments on our paper. We appreciate the time and effort you have dedicated to reviewing our work.

### Clarifications on Contributions
Our paper proposes a novel optimization framework for solving the optimization problem (P) with a regularizer parameterized by an Input-Convex Neural Network (ICNN). The key focus of our work lies in the **efficiency of optimization algorithms**, rather than improving reconstruction quality through varying the choice of regularizer.

To ensure a fair comparison, we evaluate algorithms on the same optimization problem, sharing the same objective function and regularizer in problem (P). Given the nonsmooth nature of the problem, we employ subgradient methods as a baseline, which are widely used in the literature for such problems. We acknowledge the existence of proximal gradient methods for non-smooth problems. However, as we pointed out in the paper, the proximal operator of the regularizer cannot be computed exactly. Our work circumvents the challenge with proximal operators by introducing a constrained reformulation equivalent to the original problem (Theorem 3). This is a core contribution of our framework.


### Pre-trained Regularizer
Our method assumes a pre-trained regularizer. In our experiments, we trained this in an adversarial framework, but the proposed method is independent to the training procedure. As such, we do not delve into the training process for the regularizer in this work, and it is possible that our method will lead to better reconstruction quality if the same regularizer is trained differently.

### Simplified Architecture
We intentionally employed a simple ICNN architecture to demonstrate the core effectiveness of our framework as a proof of concept. This also simplifies the ICNN training process, allowing readers to focus on our primary contributions.

### Summary of Revisions
Based on your valuable suggestions, we have revised the manuscript (revisions highlighted in blue) as follows:
1. **Introduction, Challenges, and Motivation**: We have rewritten the introduction to clearly articulate our research problem, the challenges of existing methods, and the motivations for our proposed approach. These are now detailed in Section 2.
2. **Adversarial Training and Subgradient Methods**: Additional details have been provided in the Appendix to enhance clarity.
3. **Theorem 3 Proof**: The proof has been revised for greater specificity and rigor.
4. **New Comparison**: We have included a comparison with subgradient methods applied to a smoothed version of the problem.

We hope these clarifications and revisions address the reviewers' concerns and enhance the clarity and quality of our paper. We again thank you for your valuable feedback and consideration.

---

### Meta-Review · Area_Chair_EBpv · 2024-12-15

**Metareview:**

The paper introduces a novel optimization framework for solving variational problems involving ICNN-based regularizers, which are used to parameterize convex functions. This framework addresses challenges such as the nonsmooth nature of the problem and the nested structure of ICNNs, which complicate standard optimization techniques. By reformulating the problem using epigraphical projections, the authors make it suitable for a convex optimization approach solved with a primal-dual algorithm. The paper demonstrates the efficiency of the proposed method through experiments in image reconstruction tasks such as salt-and-pepper denoising, inpainting, and computed tomography reconstruction.

Strengths: reformulation of ICNNs, solid theoretical grounding, and algorithmic efficiency

Weaknesses: limited experimental comparisons and the use of a simplistic ICNN architecture. Additional comparisons with baseline methods or applications beyond image restoration might strengthen the submission.

**Additional Comments On Reviewer Discussion:**

Despite revisions, some reviewers remained unconvinced about the paper's broader impact, particularly for ICLR's audience, and suggested that the paper might be better suited to a specialized imaging venue.

---

### Decision · Program_Chairs · 2025-01-22

Reject